# Multi-trait analysis characterizes the genetics of thyroid function and identifies causal associations with clinical implications

To date only a fraction of the genetic footprint of thyroid function has been clarified. We report a genome-wide association study meta-analysis of thyroid function in up to 271,040 individuals of European ancestry, including reference range thyrotropin (TSH), free thyroxine (FT4), free and total triiodothyronine (T3), proxies for metabolism (T3/FT4 ratio) as well as dichotomized high and low TSH levels. We revealed 259 independent significant associations for TSH (61% novel), 85 for FT4 (67% novel), and 62 novel signals for the T3 related traits. The loci explained 14.1%, 6.0%, 9.5% and 1.1% of the total variation in TSH, FT4, total T3 and free T3 concentrations, respectively. Genetic correlations indicate that TSH associated loci reflect the thyroid function determined by free T3, whereas the FT4 associations represent the thyroid hormone metabolism. Polygenic risk score and Mendelian randomization analyses showed the effects of genetically determined variation in thyroid function on various clinical outcomes, including cardiovascular risk factors and diseases, autoimmune diseases, and cancer. In conclusion, our results improve the understanding of thyroid hormone physiology and highlight the pleiotropic effects of thyroid function on various diseases.

Thyroid function tests are among the most frequently ordered biochemical tests worldwide to assess thyroid dysfunction, a common disorder with a prevalence of ~5% in the general population[1]. Thyroid hormones (TH) play a key role in the development and function of virtually all tissues during lifespan in humans[2,3]. Thyroid function is narrowly regulated by the hypothalamus-pituitary-thyroid axis[4]. The hypothalamus produces thyrotropin-releasing hormone (TRH) which acts on the pituitary gland to secrete thyrotropin (thyroid-stimulating hormone, TSH). The pituitary, in turn, stimulates the thyroid gland to produce and release TH, i.e. the prohormone thyroxine (T4) and the active hormone triiodothyronine (T3). The majority of T4 is converted intracellularly to T3 by deiodinases in peripheral tissues. T3 subsequently binds to the nuclear receptor regulating transcription of target genes. The hypothalamus-pituitary-thyroid axis is regulated by a negative feedback loop, resulting in a reciprocal physiological relationship between TSH and TH levels, aimed at maintaining adequate TH levels throughout life[5,6] (Fig. 1a).

Thyroid function is assessed by measuring circulating TSH and free T4 (FT4) levels, with, in most cases, high TSH indicating hypothyroidism and low TSH indicating hyperthyroidism. FT4 levels are decreased in overt hypothyroidism, increased in overt hyperthyroidism, and in the reference range in subclinical hypo- and hyperthyroidism. To safely guide thyroid function, reference range TSH and FT4 concentrations are used. In the last decade, it has become clear that not only overt but also subclinical hypo- and hyperthyroidism are associated with several adverse clinical outcomes, including atrial fibrillation, coronary heart disease, stroke, and mortality[7–11].

More recently, various studies have suggested that even small differences in thyroid function within the reference range are associated with clinical consequences[12,13], including increased risks of coronary heart disease, atrial fibrillation, stroke, type 2 diabetes, dementia, depression, and even mortality[14–20].

Genetic factors are responsible for an estimated 58–71% of the inter-individual variation in TSH and FT4 concentrations[21]. Over the

✉e-mail: ateumer@uni-greifswald.de; m.medici@erasmusmc.nl

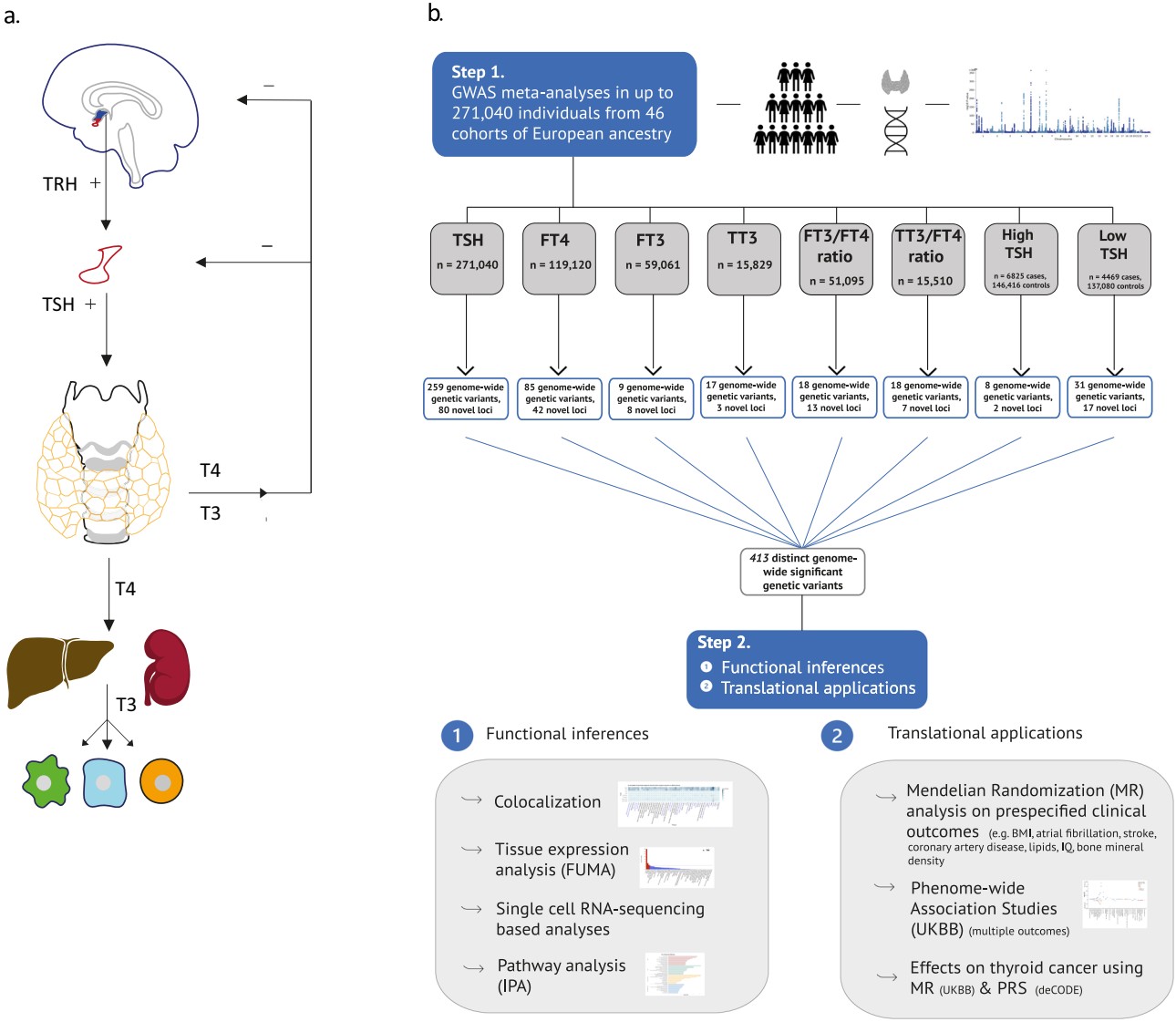

**Fig. 1 | Schematic design of the project and analyses. a** The hypothalamic-pituitary-thyroid axis is characterized by a negative feedback loop. The hypothalamus produces thyrotropin releasing hormone (TRH), which stimulates the pituitary to produce thyroxine-stimulating hormone (TSH). TSH stimulates the thyroid to produce thyroxine (T4) and triiodothyronine (T3), the active thyroid hormone affecting transcription in target cells. The majority of circulating T3 is produced by the liver and kidney by T4 to T3 conversion. **b** Step 1 represents the meta-analysis of 46 different European ancestry cohorts for eight thyroid function traits: TSH, FT4, FT3, TT3, FT3/FT4 ratio, TT3/FT4 ratio, high and low TSH. Step 2 shows the different secondary analyses performed using the meta-analyses results to identify the underlying mechanisms of the specific genome-wide variants and the translation to clinical diagnoses.

last two decades, candidate-gene and genome-wide association studies (GWAS) have identified multiple genetic variants influencing TSH and FT4 concentrations resulting in two proven new key players (*SLC17A4, AADAT*) in TH physiology[22–26]. However, the contribution of the discovered common genetic variants to the total variation in thyroid function is still limited, i.e. 9.4% and 4.8% of the total variation in TSH and FT4 concentrations, which corresponds to 33% and 21% of the variation explained by variants with a minor allele frequency (MAF) > 1%[27]. Also, common variants in known candidate genes in TH regulation such as TH transporters, metabolizing enzymes, and receptors only explain a minor part (± 2%) of the variation in TSH and FT4 concentrations, which emphasizes the need for agnostic approaches such as GWAS to further unravel the additional underlying genetic contributors[23]. Furthermore, only small studies (*n* = 577–1,731) investigated common variants affecting T3 levels and the T3/T4 ratio, a marker for TH metabolism, identifying only one genome-wide significant association with FT3 and four with FT3/FT4 ratio levels[25,27–31].

Here, we performed much larger GWAS meta-analyses on a comprehensive collection of TH traits, including TSH within the reference range, FT4, free and total T3 concentrations, and free and total T3/FT4 ratio in up to 271,040 euthyroid individuals of European ancestry from the joint collaboration of the ThyroidOmics Consortium (www.thyroidomics.com). Furthermore, we conducted case-control GWASs of high TSH versus reference range TSH and low TSH versus reference range TSH assessing thyroid dysfunction in up to 6,712 cases. As samples from other ethnicities with TH measurements were hardly available at the time we conducted this study, we restricted the analyses to European ancestry. Besides identifying many novel genetic variants, we revealed potential causal genes using colocalization analyses, various cancer and cardiac-related pathways, associations with many clinical endpoints including cardiovascular risk factors and diseases and autoimmune diseases using polygenic risk scores and Mendelian Randomization (MR), and confirmed that variation in TSH concentrations within the reference range is causally associated with

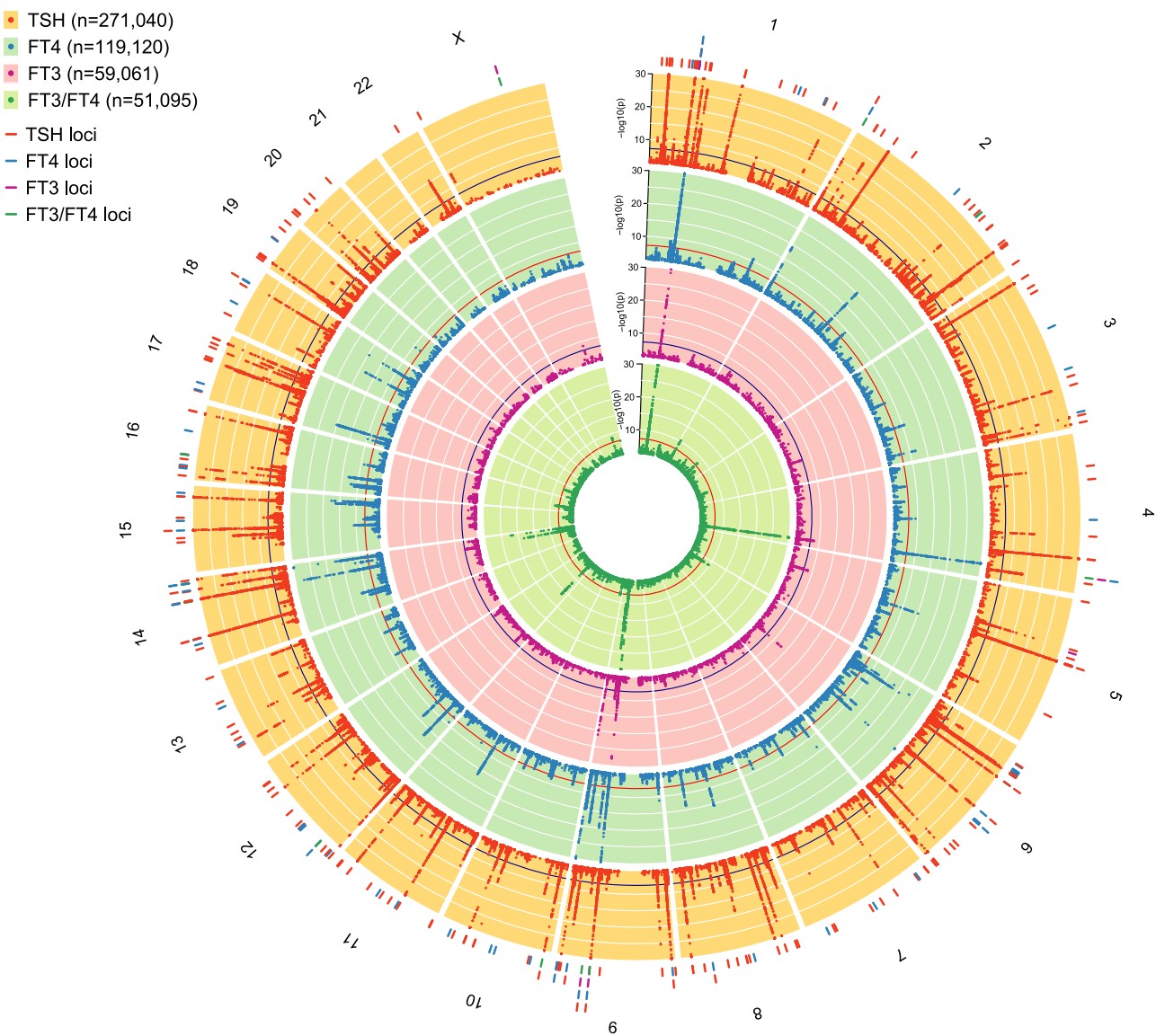

**Fig. 2 | Genome wide association results for TSH, FT4, FT3 and FT3/FT4 ratio.** The circos plot depicts the association results for TSH, FT4, FT3 and the FT3/FT4 ratio combined: red band: $-\log_{10}(p)$ for association in the meta-analysis of TSH, ordered by chromosomal position. The blue line indicates genome-wide significance ($p = 5 \times 10^{-8}$). Blue band: $-\log_{10}(p)$ for association with FT4, ordered by chromosomal position. The red line indicates genome-wide significance ($p = 5 \times 10^{-8}$). Purple band: $-\log_{10}(p)$ for association with FT3, ordered by chromosomal position. The blue line indicates genome-wide significance ($p = 5 \times 10^{-8}$). Green band: $-\log_{10}(p)$ for association with the FT3/FT4 ratio, ordered by chromosomal position. The red line indicates genome-wide significance ($p = 5 \times 10^{-8}$). The outer band indicates the positions of the associated loci as defined in Methods. Adjacent loci for a trait with the same gene names are merged. The color follows the same pattern as the association plots of the four traits. All $p$-values were obtained from two-sided association tests (z-statistics), where correction for multiple testing is indicated by the level of genome-wide significance.

thyroid cancer. An overview of the study design including the GWAS sample sizes is provided in Fig. 1b.

## Results

### Novel loci associated with thyroid function

We conducted GWAS meta-analyses of reference range thyroid function including up to 271,040 euthyroid individuals from European ancestry across 46 predominantly population-based cohorts from the ThyroidOmics Consortium. Detailed descriptions of the included cohorts with corresponding number of participating individuals per trait, age, genotyping information and quality control cut-offs are provided (Fig. 1b, Supplementary Data 1 and Supplementary Fig. 1). The LD Score Regression intercepts were between 0.99 and 1.07, supporting the absence of unaccounted population stratification that may otherwise cause inflated GWAS meta-analysis results. In total, 259 independent genetic variants were associated with TSH within the reference range and mapped into 79 known and 80 novel loci (see Methods), and 85 genetic variants with FT4 in 23 known and 42 novel loci. For the understudied thyroid traits, 9 variants (8 loci) were associated with FT3, 17 (3 loci) with TT3, 18 (13 loci) with FT3/FT4, and 18 (7 loci) with the TT3/FT4 ratio (Figs. 2 and 3 and Supplementary Figs. 2–4, Supplementary Data 2–7). For the case-control GWAS on high TSH (levels above cohort-specific reference range) and low TSH (levels below cohort-specific reference range), a subset of 27 and 20 studies with 6712 and 4212 cases (146,529 and 137,337 controls), respectively, were available for analyses. These analyses yielded 8 independent genetic variants in 6 known and 2 novel loci for high TSH and 31 variants in 8 known and 17 novel loci for low TSH (Supplementary Data 8 and 9). Descriptions of the annotated gene for each locus with respect to its function, expression and role in monogenic

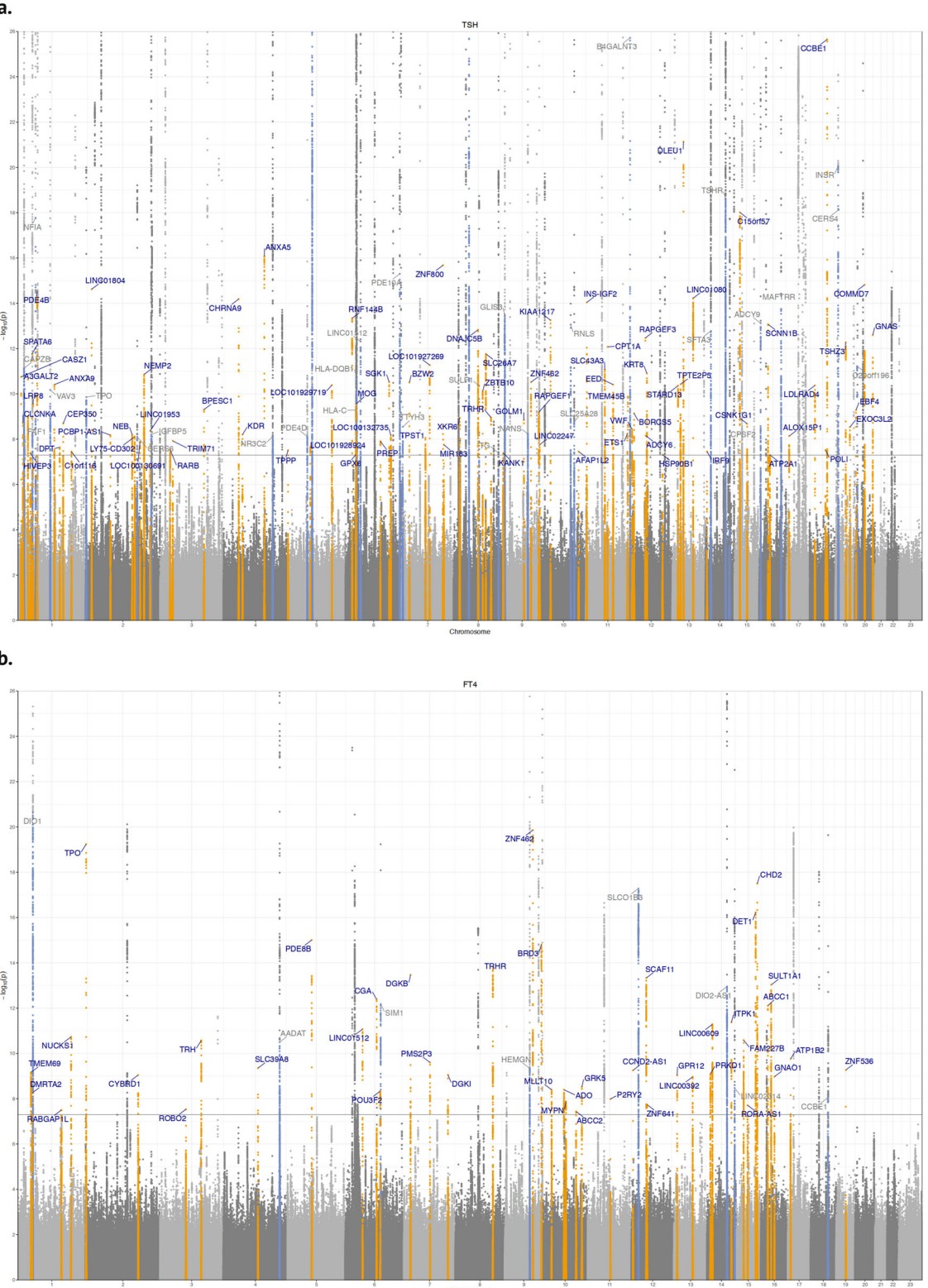

**Fig. 3 | Zoomed Manhattan plot for TSH and FT4.** Zoomed Manhattan plot of the GWAS meta-analysis results for TSH (panel **a**) and FT4 (panel **b**). Variants are plotted on the x-axis according to their position on each chromosome with the -$\log_{10}$(p-value) of the association test on the y-axis. The horizontal line indicates the threshold for genome-wide significance, ($p = 5 \times 10^{-8}$). All p-values were obtained from two-sided association tests (z-statistics), where correction for multiple testing is indicated by the level of genome-wide significance. Novel loci are colored in orange, and novel independent associations within known loci are colored in light blue. Genetic variants were assigned to the nearest gene. Variants were considered known when they are in linkage disequilibrium with a previously identified variant (see Methods).

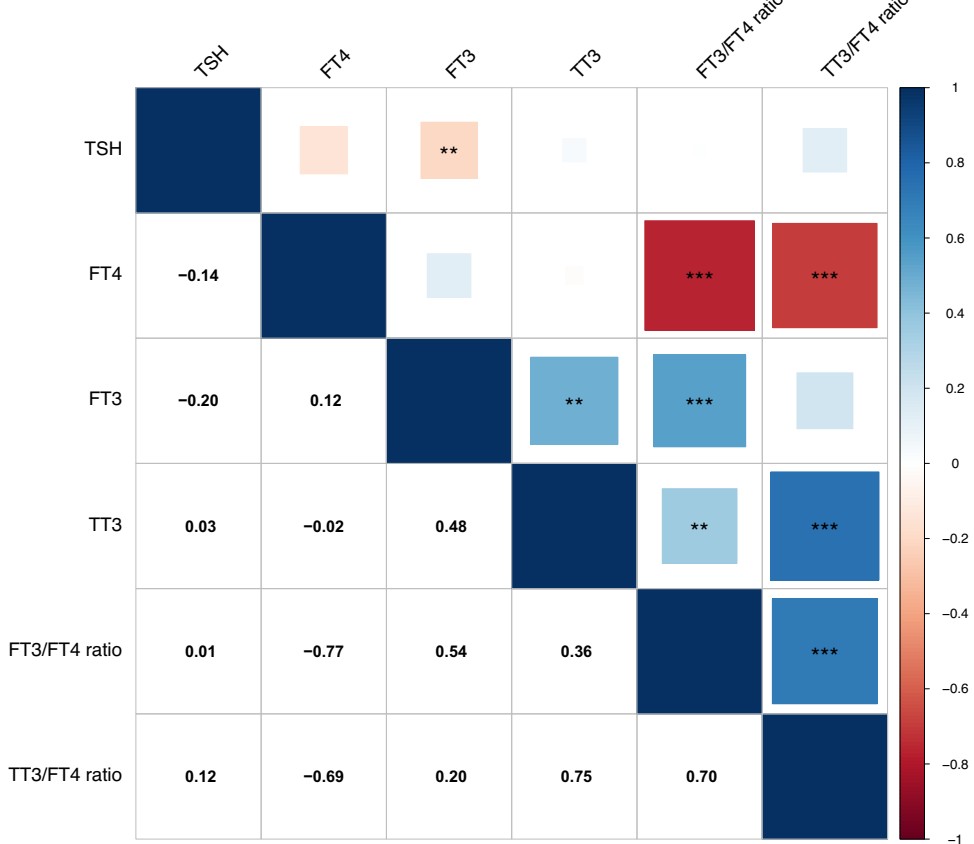

**Fig. 4 | Genetic correlations of thyroid hormone parameters.** Pairwise genetic correlations were estimated via bivariate LD score regression. In the upper part, positive genetic correlations are shown in blue, and negative correlations are depicted in red as indicated by the legend. The lower part shows the genetic correlation values. FDR was calculated via the Benjamini–Hochberg method to correct for multiple testing of all 15 correlations. Larger squares correspond to stronger genetic correlation. Significant correlations are indicated by asterisks (FDR: * <0.05, ** <0.001, *** <0.0001).

diseases can be found in Supplementary Data 10. The majority of the associations showed no to moderate heterogeneity (I² statistics) for the continuous traits, whereas heterogeneity was generally higher for the two binary trait GWAS (Supplementary Data 2–9). Although no independent sample for replication was available, all significant associations for TSH and FT4 had the same effect directions in former GWAS supporting the robustness of the results (Supplementary Fig. 5, Supplementary Data 11). Quantile-quantile plots are shown in Supplementary Fig. 6. The conditional analyses using GCTA as an alternative approach to assess independent secondary signals within a locus showed a strong concordance with the clumping results (Supplementary Data 12).

Fine-mapping revealed credible sets with sizes between one and 1149 variants, where the most sets with five or less variants were found for TSH (*n* = 111) and FT4 (*n* = 35) (Supplementary Data 13, Supplementary Fig. 7a, b). Although this was not surprising given that these two traits had the most associated loci in total, there was no significant difference on the average credible set size among the trait-associated GWAS loci that would indicate an enrichment of potential causal variants (*p*-value$_{ANOVA}$ = 0.97). For TSH, exonic variants with posterior inclusion probabilities >0.8 were found in *ZNF800*, *B4GALNT3*, *TSHR*, *SCNN1A*, and *TG* (Supplementary Fig. 7c), and in *SLC39A8* for FT4 (Supplementary Fig. 7d), but none for the other continuous traits.

Limited overlap in terms of a similar SNP (rsid) between reference range TSH and FT4 genome-wide significant variants was observed, namely only for rs954585 in the *ZNF462* gene, rs116909374 in *MBIP* and rs17477923 in *FAM227B*. These three variants were also significant in the low TSH GWAS. Moreover, two other low TSH variants showed overlap

with TSH: rs10799824 in *CAPZB*, and rs114322847 in *TG*. Two variants showed overlap between FT4 and FT3, namely rs2235544 in *DIO1* with opposite effect directions for FT4 and FT3, and rs4842131 in *LHX3*, with similar effect directions for FT4 and FT3. This is consistent with expected underlying physiological mechanisms, as for example the rs2235544 variant has been associated with less *DIO1* activity, thereby resulting in higher FT4 and lower FT3 concentrations[32]. Several previously reported associations with TSH and FT4 were also genome-wide significant in our meta-analyses and located in genes known to be of importance in TH synthesis *(GLIS3, TPO, TG)*, TSH synthesis and signaling (*LHX3, CGA, TSHR, PDE8B*), TH metabolism *(DIO1, DIO2, AADAT)* and transport *(SLCO1A2, SLCO1B3, SLC17A4),* while also numerous genes of unknown importance were identified (Supplementary Data 2 and 3). We calculated the explained variance (h² ± SE) by all common and low-frequency variants with a minor allele frequency (MAF) > 1% of the total variation in thyroid related traits yielding: 30.7 ± 8.6% (TSH), 23.3 ± 8.6% (FT4), 21.3 ± 8.6% (FT3), 42.6 ± 37.9% (TT3), 25.1 ± 8.5% (FT3/FT4 ratio) and 10.1 ± 38.6 (TT3/FT4 ratio). The discovered genome-wide significant variants together explained 14.1% (*n* = 259 variants) and 6.0% (*n* = 85 variants) of the total variation in TSH and FT4 serum concentrations, respectively. For the T3 related traits we explained 1.1% (FT3, *n* = 9 variants), 9.5% (TT3, *n* = 17 variants), 2.7% (FT3/FT4 ratio, *n* = 18 variants) and 7.0% (TT3/FT4 ratio, *n* = 18 variants). Based on the significant genetic correlation between TSH and FT3 (genetic correlation = −0.2, FDR < 0.001), the associated loci of these traits seem to reflect thyroid function determined by the active thyroid hormone FT3. The non-significant correlation of FT4 with TSH in combination with the significant correlation of FT4 with both ratios (all FDR < 0.001) suggest that

a.

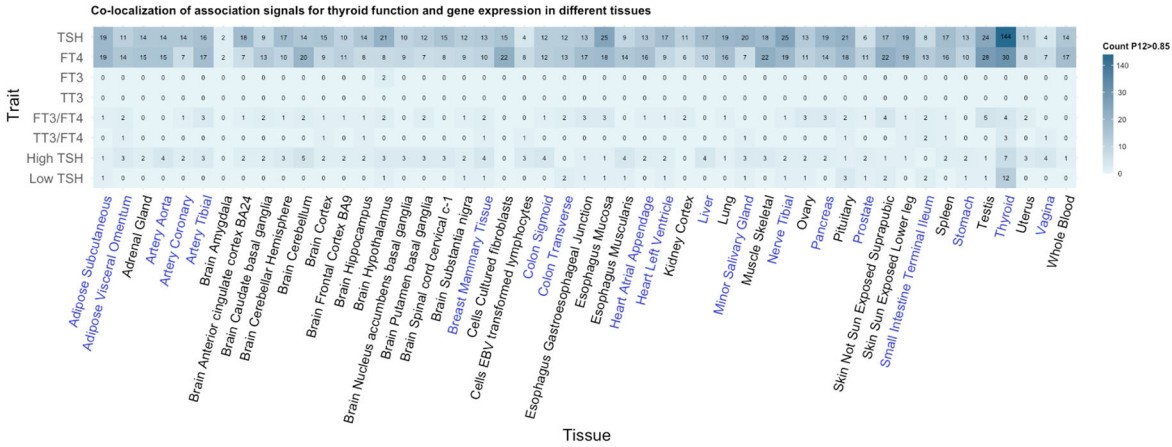

b.

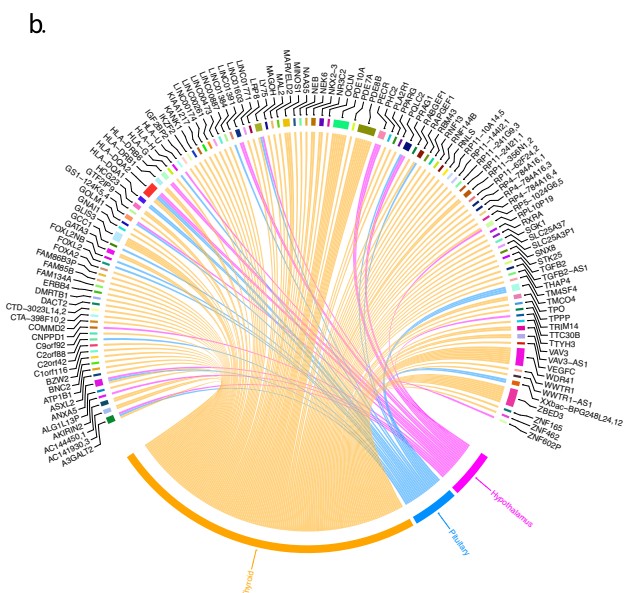

c.

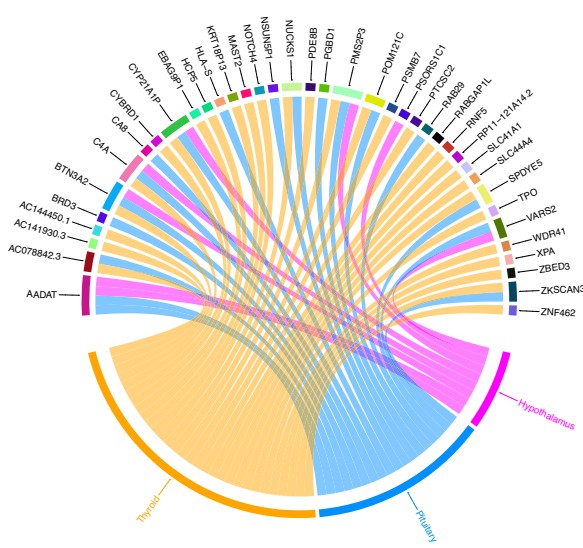

**Fig. 5 | Colocalization of associations for thyroid function parameters and gene expression.** In panel **a**, the different thyroid function traits are shown on the y-axis and the tested tissues (*n* = 49) derived from the GTEx database are shown on the x-axis. The number of significant colocalizations between thyroid function genome-wide significant variants and gene expression in the different tissues are shown in each box using a probability (P12) > 0.85 to confirm the H4 hypothesis (same shared causal variant). Tissue names are similarly colored when present in the same organ or belonging to the same group of tissues (blue or black). Detailed results of the colocalizations of TSH (panel **b**) and FT4 (panel **c**) are shown in the tissues of the hypothalamus-pituitary-thyroid axis.

the FT4 associated loci reflect the thyroid hormone metabolism assessed via the T3/T4 ratio. The significant genetic correlation of TT3 with FT3 (genetic correlation = 0.5, FDR = 0.005) indicates that also the TT3-associated loci predominantly reflect the active thyroid hormone (Fig. 4). Taking into account the strong effects of the *SERPINA7* locus for TT3, this could explain the non-significant genetic correlation of TT3 with TSH.

## Colocalization

To assess whether the effect of a genetic variant on the thyroid trait may operate via differential gene expression, colocalization analysis was performed using GWAS results of thyroid function traits and expression quantitative trait loci (eQTL). In total, available eQTLs over 49 different tissues were selected from the GTEx Project database. (https://gtexportal.org/). All findings that showed colocalization (posterior probability of a shared underlying causal variant >0.85,

Methods) are provided in Supplementary Data 14. For TSH associated variants, we identified 830 colocalizations for 237 different mRNA transcripts, predominantly in thyroid tissue (*n* = 144 positive colocalizations) (Fig. 5), and few in hypothalamus and pituitary (both *n* = 21 positive colocalizations). These included multiple genes with a known role in the TSH signaling cascade and TH synthesis, such as *PDE8B*, *PDE10A*, *TPO* and *GLIS3*. For FT4, we identified 630 positive colocalizations for 146 different mRNA transcripts in various peripheral tissues. Amongst others, these included *AADAT* in the small intestine and adipose subcutaneous tissue (rs76767373, rs112649654), which is a known TH metabolizing enzyme. TT3 levels, the ratios, and high as well as low TSH showed few to no colocalizations based on *cis* eQTLs in the examined tissues, likely due to lower statistical power to detect positive colocalizations. Only one variant associated with FT3 concentrations, rs17628883, influenced *AADAT* transcript levels in the hypothalamus. A detailed visualization of genes colocalized in tissues

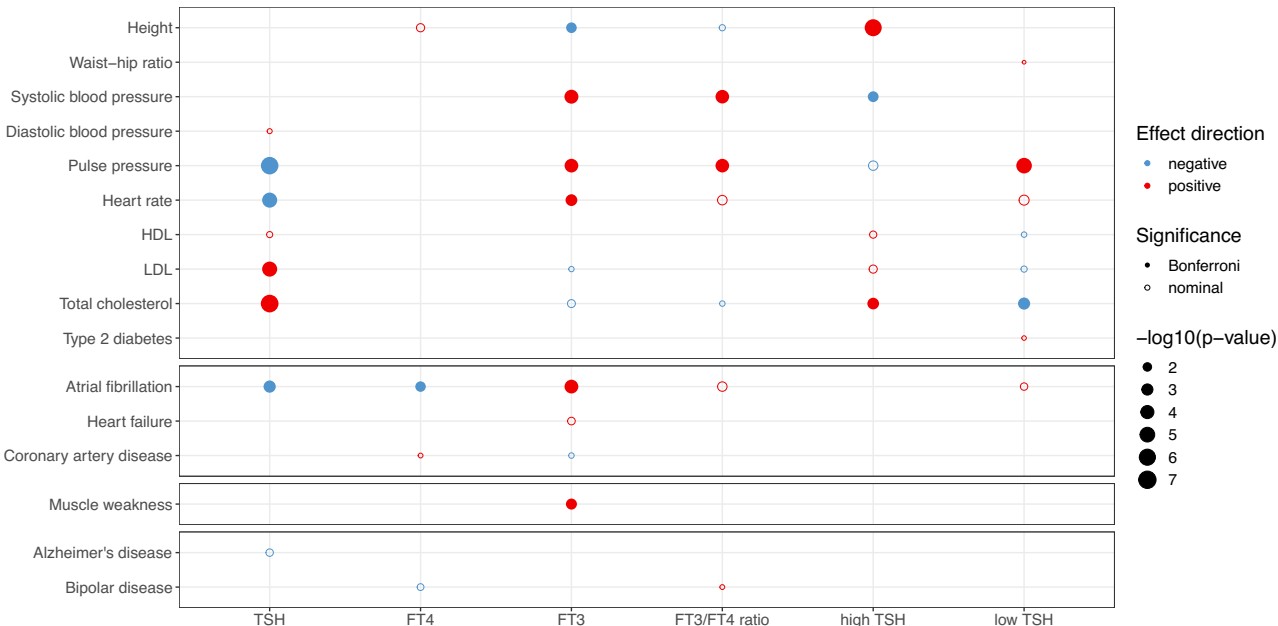

**Fig. 6 | Effects of genetic variants on different thyroid-related outcomes using Mendelian Randomization.** Twenty-four clinical outcomes were tested against the thyroid-related parameters. Significance thresholds are depicted either with a closed bubble (Bonferroni corrected) or an open bubble (nominal significance at $p < 0.05$). The direction of effect is depicted in color: blue shows a negative beta and red is a positive beta. The size of the bubble indicates the degree of the association in terms of -$\log_{10}(p$-value). All $p$-values were obtained from two-sided weighted median Mendelian Randomization tests. Clinical outcomes with at least one association with thyroid function parameters are depicted. Bone mineral density, anxiety, and major depressive disorder with zero associations are therefore not shown.

of the hypothalamus-pituitary-thyroid axis is provided in Fig. 5b, c, and in Supplementary Fig. 8. It can be seen that genes like *PDE8B* and *TPO* involved TSH signaling cascade and TH synthesis colocalized in thyroid tissue only.

**Functional enrichment and pathway analyses**

Using DEPICT, genes were assigned to GWAS results of traits with >10 associations in autosomal regions (Supplementary Data 15) and subsequently used as input for the Ingenuity Pathway Analysis tool. The top ten enriched canonical pathways for the TSH, FT4, FT3/FT4 ratio, and low TSH are illustrated in Supplementary Fig. 9[33]. The representative genes in each of these pathways are listed in Supplementary Data 16. The pathway with the lowest $p$-value for association with TSH and low TSH was relaxin signaling. Relaxin is a pleiotropic hormone mediating hemodynamic changes in different tissues[34]. Related peptides influence the secretion of hormones by the pituitary gland including TSH, thereby promoting thyroid growth as supported by early animal and in vitro studies, while also involvement in thyroid carcinogenesis has been suggested[35–39]. Other pathways were cardiac hypertrophy signaling and cardiac beta-adrenergic signaling, which is of interest given the known effects of TH on pulse rate, the risk of rhythm disorders and heart failure via facilitation of beta-adrenergic signaling[40]. For FT4 and the FT3/FT4 ratio, significantly associated pathways *(DIO1 and DIO2 genes)* were related to TH metabolism and biosynthesis, consistent with expectation and underscores the plausibility of these results. In addition, tissue enrichment analyses were performed using FUMA[41]. After multiple testing correction for the number of tissues, significant enrichment was seen only for TSH variants, with thyroid tissue showing the strongest association, followed by the stomach and prostate (Supplementary Fig. 10).

**Single-cell RNA based analyses**

To characterize putative causal cell types responsible for variation in thyroid function parameters within the reference range, single-cell RNA data of healthy thyroid tissue were used[42]. Of the investigated cell types (see Methods), genes identified in the TSH and low TSH GWASs were significantly enriched in thyroid epithelial cells (TSH $p$-value = 0.0004, low TSH $p$-value = 0.0009). No associations with other cell types were observed for the aforementioned and other tested thyroid function-related traits (Supplementary Data 17).

**Causal associations with thyroid function-related outcomes**

Frequent and clinically relevant outcomes related to thyroid (dys) function (e.g. anthropometric, cardiovascular risk factors and endpoints) were selected and causal associations with thyroid function were assessed using Mendelian Randomization. All traits including details of the underlying samples are provided in Supplementary Data 18, and all associations of the instruments with the outcomes are given in Supplementary Data 19. Increased TSH levels within the reference range were causally associated after correction for multiple testing (see Methods) with a less favorable lipid profile (higher LDL-cholesterol and total cholesterol), a lower heart rate and pulse pressure, and a lower risk of atrial fibrillation (Fig. 6, Supplementary Data 20). FT4 was causally associated with a lower risk of atrial fibrillation whereas FT3 showed causal associations with a higher risk of atrial fibrillation, systolic blood pressure (SBP), pulse pressure, heart rate and muscle weakness, and lower height. The findings of high TSH for total cholesterol overlapped with the results of reference range TSH, as well as for total cholesterol and pulse pressure with low TSH (but with expected opposite effect directions). The FT3/FT4 ratio showed similar effect directions as FT3, and thus opposite to TSH for pulse pressure. Waist-hip ratio, triglycerides, bone mineral density, fracture risk, stroke, major depressive disorders, anxiety, and intelligence showed no nominally significant causal associations ($p$-value ≥ 0.05) with any thyroid function parameter.

**Pleiotropic effects of thyroid function parameters on clinical endpoints**

In the phenome-wide association study using polygenic scores (PGS), associations were found between reference range TSH-increasing

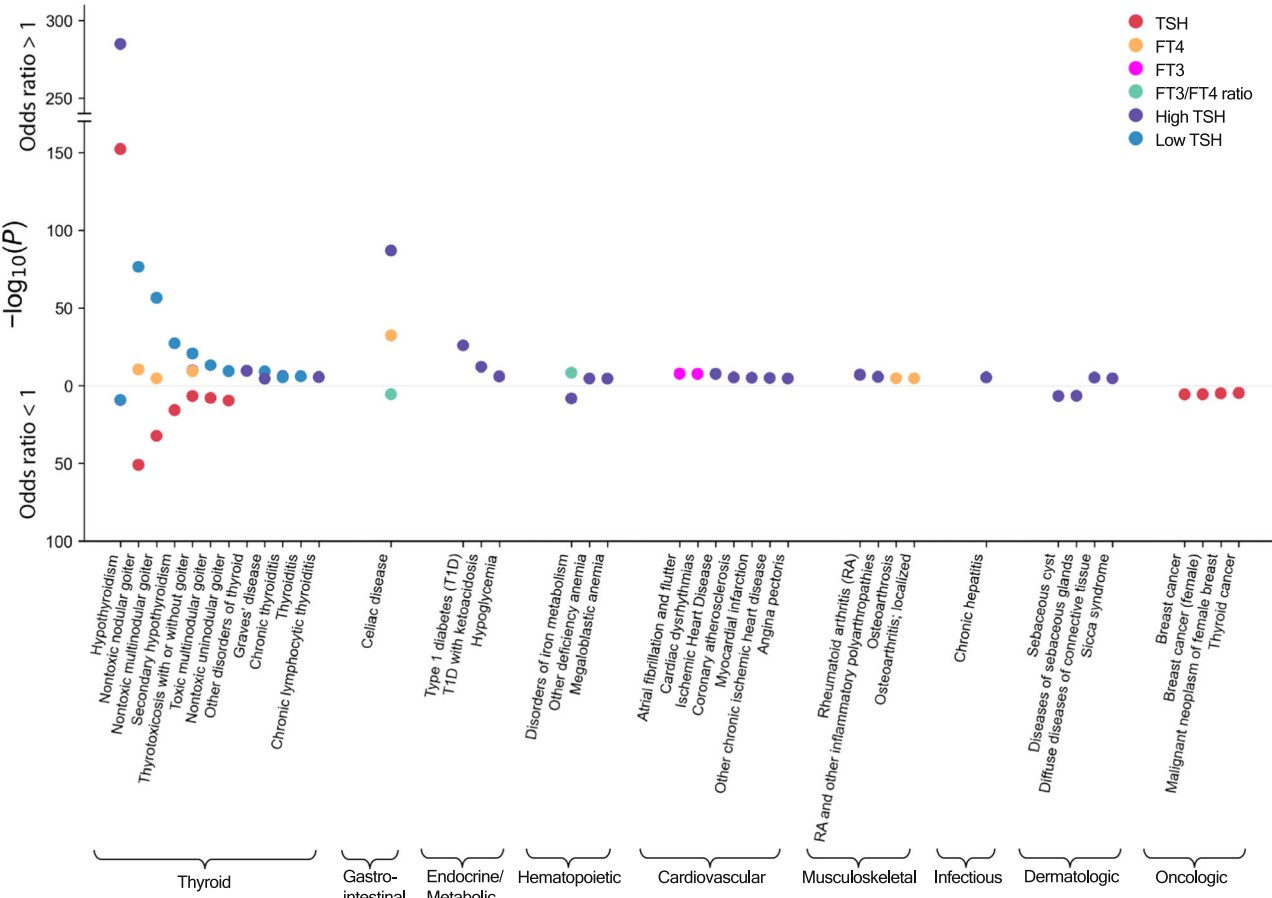

**Fig. 7 | Polygenic risk score association results.** Significant associations between thyroid function parameter polygenic scores (PGSs) and diseases in the UKBB. The data points are color-coded by trait (legend) by phenotype groups (x-axis) and -log₁₀(p-value) obtained from two-sided association tests (z-statistics) together with the direction of the effect (y-axis). All results shown passed the Bonferroni significance threshold ($p < 0.05/1460$).

alleles and an increased risk of hypothyroidism, whereas opposite effect directions were seen for FT3 (Fig. 7, Supplementary Data 21). Also, the PGS for TSH was associated with a lower risk of secondary hypothyroidism and (non)toxic (multi)nodular goiter. The PGS for FT4 was associated with a higher risk of thyrotoxicosis. Genetic associations with cardiac diseases were observed for increased FT3 concentrations demonstrating an increased risk of cardiac dysrhythmias, in particular atrial fibrillation and flutter, while genetic predisposition to high TSH levels was associated with a higher risk of coronary atherosclerosis, angina pectoris, ischemic heart disease, and myocardial infarction. Furthermore, the results suggested that thyroid function-related genetic variants, particularly the variants associated with high TSH levels, also play a role in the development of autoimmune disorders, including autoimmune thyroid disease (Graves' disease and chronic lymphocytic thyroiditis also known as Hashimoto's disease), celiac disease, rheumatoid arthritis, type 1 diabetes mellitus, and sicca (Sjögren) syndrome. The PGS for reference range TSH was also associated with a lower risk of thyroid and breast cancers.

### Assessment of the relation with thyroid cancer
Each higher PGS quartile of TSH increasing alleles was significantly associated with a lower risk of thyroid cancer (n = 620, OR = 0.82, OR confidence interval [CI] 0.77–0.88, p-value = 1.0 × 10⁻⁷), which is in agreement with previous reports[24] (Fig. 8, Supplementary Data 22). In line with this finding, the PGS obtained from the high and low TSH GWAS were also significantly associated with thyroid cancer with concordant effect directions (Supplementary Fig. 11). Furthermore, the FT3-based PGS was significantly associated with an increased risk of

thyroid cancer (n = 686, per quartile OR = 1.16, CI 1.08–1.24, p-value = 2.90 × 10⁻⁰⁵) (Fig. 8), while this was not the case for FT4 (Supplementary Fig. 11a). The results of the MR analyses supported a causal association between TSH and a lower risk of thyroid cancer (IVW: β = −0.58, p-value = 1.3 × 10⁻⁰⁷; Supplementary Figs. 12 and 13). There was no evidence for a causal association between the other thyroid function parameters and thyroid cancer (Supplementary Data 23 and 24).

## Discussion
Our study included 46 studies collaborating in the ThyroidOmics Consortium, a platform committed to improve the understanding of the genetic basis underlying thyroid function and diseases. We substantially expanded previous knowledge on the genetic basis of TSH and FT4, by increasing the number of novel loci with 74% for reference range TSH and 115% for FT4, and revealing novel variants associated with FT3, TT3, and T3/FT4 ratios. Next to TSH, FT4, high TSH and low TSH we also included free T3, total T3, and T3/FT4 ratios as markers for TH metabolism. The independent genome-wide significantly associated TSH and FT4 variants explain 14.1% and 6% of the total variation in TSH and FT4, respectively, which is an increase of 50% and 25% compared to previous GWASs on these traits in individuals with reference range TSH levels[23,27]. We revealed novel variants associated with FT3, TT3, and/or T3/FT4 ratios, providing a starting point for subsequent functional analyses and phenotyping. Furthermore, we explained with only few variants a large part of the variation in TT3 (9.5%). This is mainly driven by a common missense variant in the *SERPINA7* gene (rs1804495; p.L303F) on chromosome X, which

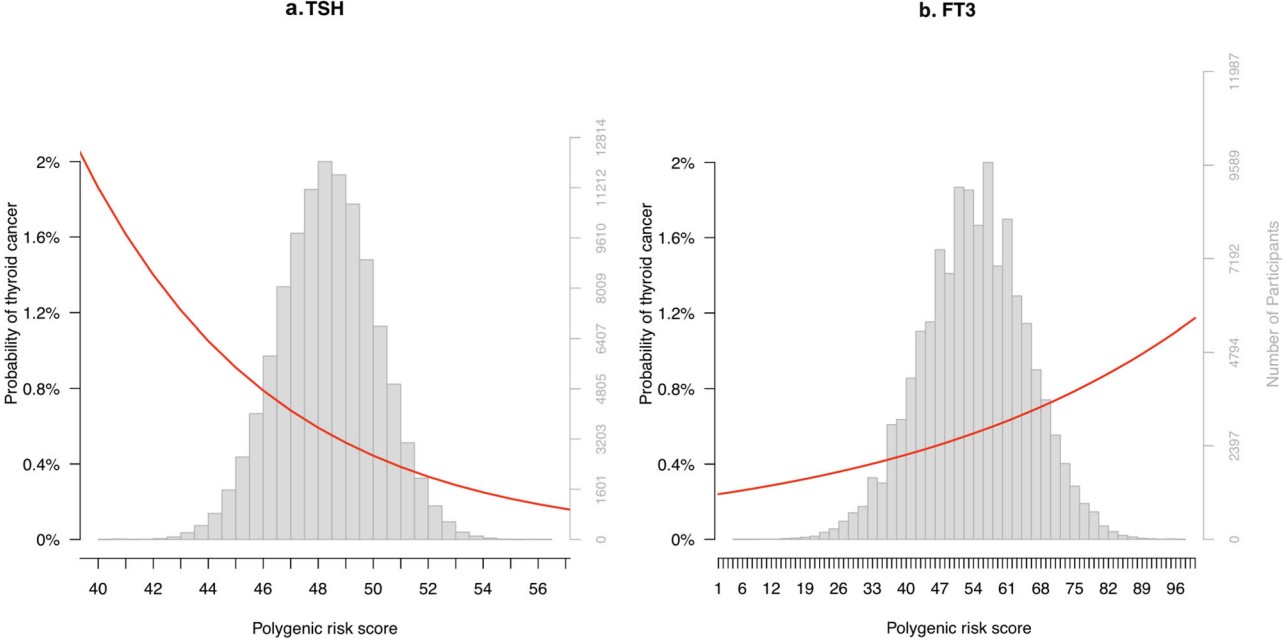

**Fig. 8 | Thyroid function polygenic risk scores and the risk of thyroid cancer.**
Plots of the associations between thyroid parameter polygenic scores TSH (panel **a**) and FT3 (panel **b**) and thyroid cancer in deCODE (**a**; $N_{case}$ = 620, $N_{control}$ = 106,168, **b**; $N_{case}$ = 686, $N_{control}$ = 119,187). The y-axis shows the probability of thyroid cancer.

The x-axis shows the percentage of risk alleles carried out based on a weighted polygenic score. The histogram shows the distribution of the polygenic score in the study sample per trait. $N_{case}$: sample size of cases. $N_{control}$: sample size of controls.

encodes thyroxine-binding globulin (TBG), one of the serum T4 and T3 binding proteins[43]. We successfully replicated previously known loci from the meta-GWAS on reference range thyroid function of the ThyroidOmics Consortium and almost all (98 out of 99) of the genome-wide significant variants identified in a whole range TSH GWAS[24,27]. We did not replicate the genome-wide significant TSH variant rs13037502 (*PTPN1*, C allele, $MAF_{average}$ = 0.10, $MAF_{Croatian}$ = 0.18) and FT3 variant rs67142165 (*EPHB2*, T allele $MAF_{average}$ = 0.04, $MAF_{Croatian}$ = 0.13) found in a small Croatian GWAS ($n$ = 1,731), which may be attributed to either population-specific variants or false positivity[25].

In addition to TH pathway genes also identified in previous GWASs, our meta-analyses revealed common variants in several genes with an established role in TH synthesis, transport, and metabolism, which have not previously been detected in GWASs. These included variants in: *TRH* (FT4 GWAS); *TRHR*, encoding the TRH receptor (TSH and FT4 GWAS); *CGA*, encoding the alpha chain of TSH (FT4 GWAS); *SLC26A7*, an iodide transporter important in thyroid hormonogenesis (TSH GWAS); *SERPINA7*, encoding TBG (FT3, TT3 and FT3/FT4 ratio GWAS), and *SULT1A1* (FT4 GWAS) and *SULT1A2* (FT3/FT4 ratio GWAS), which are sulfotransferases responsible for TH conjugation. Taken together, these results confirm that GWASs on thyroid function are able to identify loci that contain biologically plausible genes involved in TH regulation. This is of particular value, because it implies that the novel loci uncovered in this GWAS could contain yet unknown players in TH regulation. This is of interest as two novel players, *AADAT* encoding a TH metabolizing enzyme and *SLC17A4* encoding a TH transporter, have been identified via GWAS[27]. Furthermore, tissue expression analyses for the TSH-associated loci showed an enrichment of genes expressed in the thyroid. Next, single-cell RNA sequencing analyses demonstrated that among various cell types present in the thyroid, the epithelial cell (i.e., the thyrocyte) is the responsible cell type for the production and secretion of T3 and T4. Indeed, the thyrocyte expresses the TSH-receptor, and the subsequent intracellular TSH signaling pathways stimulate thyroid hormone production (Supplementary Data 17).

Considering the inverse association between TSH and FT4 in the hypothalamus-pituitary-thyroid axis, one might have expected more overlap in variants implicated in the variation of both hormones (Fig. 4). Our analyses were most powerful for TSH as a continuous trait restricted to the reference range. The relationship between circulating TSH and FT4 levels means that a certain change in TSH levels is likely to be associated with an even smaller or indeed no detectable change in FT4 levels[6,44]. In line with this, tissue expression analyses (Supplementary Fig. 10) showed that the genes implicated by the TSH GWAS were predominantly expressed in thyroid, whereas the FT4 GWAS pointed to genes that are highly expressed in various peripheral tissues and included genes determining peripheral TH bioavailability, such as the *DIO1*, *DIO2*, and *AADAT* genes encoding TH metabolizing enzymes. This latter finding is also reflected in FT4 associated loci showing higher genetic correlation with thyroid hormone metabolism (T3/FT4 ratios. FDR < 0.001) compared to the correlation with TSH (Fig. 4).

Several autoimmune diseases, including autoimmune thyroid diseases, cosegregate[45]. Our PheWAS showed associations of genetically predicted thyroid function with autoimmune diseases such as celiac disease, sicca syndrome, type 1 diabetes mellitus and rheumatoid arthritis. We identified that variants in *PSORS1C1* and *HLA-DQA1* were associated with high TSH. These genes have been implicated in one or more other autoimmune diseases[46–49]. Also, all eight high TSH variants were significant variants in a published autoimmune thyroid disease GWAS including both hypothyroidism and Graves' disease cases (Supplementary Data 25)[50]. Interestingly, genetic variation in various well-known autoimmune genes (*HLA-C, HLA-DQA1, HLA-DQB1, SH2B3, STAT4*) was also associated with variation in reference range TSH levels[51,52]. This raises the question of whether population-based TSH reference ranges are truly representative for all individuals, supporting the view that every individual has a unique setpoint located within these wide population-based reference ranges[53]. Furthermore, we included chromosome X in our analyses, which plays an important role in development of autoimmune diseases[54].

MR analyses were performed to obtain a complete overview of the causal effects of genetically determined variations in thyroid function

parameters on the risk of the most frequent and clinically relevant outcomes (Fig. 7). Although the novel set of instruments for TSH did not replicate the causal association with stroke, these analyses confirmed previous MR findings that reference range thyroid function has pleiotropic effects: genetically determined higher TSH levels (within the reference range) were associated with a less favorable lipid profile and a lower risk of atrial fibrillation[55–57]. We additionally showed that genetically determined higher TSH levels (within the reference range) were associated with a lower heart rate and pulse pressure. This study is of additional value as we also included FT3-associated genetic variants in MR. We detected more overlap of similar causal associations between TSH and FT3 than between TSH and FT4, which is in line with the higher genetic correlation between TSH and FT3 compared to TSH and FT4 (Fig. 4). For the first time, genetically determined FT4 and FT3 showed causal associations with a lower risk (FT4) and higher risk (FT3) of atrial fibrillation ($p$-value < 0.002) and higher (FT4) and lower risk (FT3) of coronary artery disease ($p$-value < 0.05) respectively[56–59]. Of interest, the most significant FT3 hit (*DIO1*-rs2235544) is also the most significant FT4 hit and has in line with known physiology opposite effects on FT3 and FT4 levels[32]. Opposite effects were also observed for two other significant FT3 instruments which were in LD with significant FT4 hits: *PTCSC2*-rs1588635 (EAF$_{A\text{-allele}}$ = 0.34) vs FT4-hit rs965513 (EAF$_{A\text{-allele}}$ = 0.34, $r^2$ = 1.0, D' = 1.0) and *ZNF462*-rs7033661 (EAF$_{A\text{-allele}}$ = 0.64) vs FT4-hit rs954585 (EAF$_{A\text{-allele}}$ = 0.64, $r^2$ = 1.0, D' = 1.0) (Supplementary Data 3–4). Such effects of genes involved in TH metabolism could partly explain the observed opposite effects of FT3 and FT4 instruments on clinical outcomes.

In population-based as well as MR studies, thyroid function and thyroid diseases have been associated with an altered risk of several solid cancers[60–64]. In the current study, pathway as well as PheWAS analyses highlighted the involvement of thyroid function in different types of cancers, such as thyroid, breast, ovarian and colorectal cancer. While causal associations between TSH and ovarian or colorectal cancers have not been confirmed via MR previously, we found higher genetically predicted TSH associated with a lower risk of breast and thyroid cancer which is in line with current literature[24,64]. The detected causal association between higher TSH levels and a lower thyroid cancer risk seems counterintuitive, as TSH is known to promote thyroid growth. Yet, as we have shown in this study, multiple genetic variants associated with TSH concentrations are influencing genes that are expressed in the thyroid which could alter the sensitivity of the thyroid to TSH, thereby leading to higher TSH levels. On the other hand, some of the genes determining variation in TSH levels are also known (oncogenic) genes in thyroid cancer such as *MBIP, IGFBP5* and *B4GALNT3*[65–67]. The net effect of these various mechanisms seems to result in a lower risk of developing thyroid cancer, which should be further substantiated in future studies.

While our study doubled the sample size for TSH and FT4 with respect to previous studies, this study was limited to common and low frequency polymorphisms (MAF > 1%). Rare variants (MAF < 1%) might also contribute to a certain extent of the variation in thyroid function parameters in addition to the well-known environmental factors (e.g., age, sex, body mass index, smoking and iodine status.). For this, large exome or whole-genome sequencing studies are required to explain the missing heritability. The calculated $h^2$ for TT3 and TT3/FT4 ratio included large SEs which could well be due to the low sample size ($n$ = 842) of the NBS study in which the explained variance for these two traits was estimated, in contrast to $n$ = 3326 individuals of the SHIP-START cohort used for the $h^2$ estimation of the remaining traits. Also, at the time of conducting this study, no population-based studies other than from European ancestry including TH measurements were available. Therefore, the findings cannot be extrapolated to other ancestries. Moreover, pathway analyses also have limitations as results rely on annotated genes combining different sources with variable coverage.

In conclusion, we conducted large GWAS meta-analyses on thyroid function including all important thyroid function parameters. In addition to increasing the number of loci discovered, this study improved our understanding of the genes altering mRNA expression in different tissues, and their contributing effects in various pathways influenced by thyroid function parameters. Furthermore, it provides a comprehensive overview of the effects of genetically determined variation in thyroid function on many known and newly suggested clinical outcomes. Given the associations of thyroid function within the reference range with various risk factors and diseases, we hypothesize that thyroid function might be better interpreted on a continuous scale, rather than a binary interpretation based on fixed reference ranges. Taken together, this study serves as an important basis for follow-up studies including in vitro studies to reveal the functional relevance of the genes, the potential for druggable targets using the results of the colocalization analysis, and further PGS and MR studies to test causal effects of thyroid (dys)function on other diseases. This could foster possibilities for using genetics in prevention and diagnosis, and identify candidates for therapeutic targets to reduce the burden of thyroid diseases.

## Methods

### Study population

Cohorts with participant data of European ancestry, collaborating in the ThyroidOmics Consortium, were asked to participate. Participants aged <18 years, of non-European ancestry, using thyroid medication (defined as ATC (Anatomical Therapeutic Chemical) code H03), or with a history of thyroid surgery were excluded from all analyses. Information regarding gender distribution, mean age, and thyroid hormone parameter measurements was gathered for all cohorts (Supplementary Data 1). Only studies having at least 40 cases were considered in the high and low TSH analyses and individuals with reference range TSH were used as controls. The final sample sizes for the GWASs are presented in Fig. 1b. Each participating study was approved by the respective ethics committee, and all participants provided written informed consent. Details are provided in the Supplementary Note. All inclusion criteria of the participants including sex and ethnicity are provided in the Methods and Supplementary Data.

### Trait definition and statistical analyses

Subjects with TSH concentrations within the cohort-specific reference range were included for the TSH, FT4, FT3, TT3, FT3/FT4 ratio and TT3/FT4 ratio analyses[68]. Reference ranges are usually calculated using an upper and lower 2.5% percentile of the TSH distribution thereby taking assay and population characteristics, as well as additional environmental factors like population iodine supply into account[69]. If this information was not available, the reference range for TSH provided by the assay manufacturer was used. TSH, FT4, FT3 and TT3 were analyzed as continuous variables after inverse normal transformation. Ratios were natural log-transformed prior to analyses. Individuals with TSH concentrations above the upper limit or below the lower limit of the cohort-specific reference range were defined as cases for the high and low TSH GWAS, respectively, and individuals with TSH concentrations within the cohort-specific reference range were included as controls. Detailed information of the thyroid hormone measurement assays is given in Supplementary Data 1.

### GWAS in individual studies

Genotyping on genome-wide arrays was performed in all studies. Genome-wide data were imputed to HRC version 1.1, or 1000 Genomes phase 1 or 3. In each study, a multiple linear regression model with additive genetic effect was applied to test for phenotype–genotype association for the continuous phenotypes, and logistic regression for the dichotomous traits, adjusted for sex, age and age[2] (to account for non-linear effects) and relevant cohort-specific covariates as

appropriate. These covariates include principal components for population stratification, family structure in case of family-based studies, study site, village, field center or laboratory batch. Study-specific information regarding genotyping and imputation are displayed in Supplementary Data 1.

## Meta-analyses in discovery cohorts

All result files from the included studies underwent extensive quality control (QC) checks. File-level QC included format checks, such as the removal of duplicate variants or associations with missing or invalid values and the harmonizing of the alleles of the INDELs. The QC of the individual study GWAS results was performed using the EasyQC package version 18.1 in R. P-Z plots were generated to detect analytical problems related to the study-specific computation of the beta-estimates (z-statistic) compared to reported $p$-values. Effect allele frequency (EAF) plots identified possible strand issues, allele miscoding or inclusion of individuals whose self-reported ancestry did not match their genetic ancestry based on reported effect allele frequencies against a reference set (HRC 1v1 and 1000 Genomes 1 phase 3). Furthermore, SE-N plots using EasyQC output were generated (sqrt(N_total) versus c_trait_transf/SE_Q0.5) to identify improper transformations of the traits, incorrect use of the regression model or unit errors. Known associations for TSH (rs6885099 in *PDE8B)* and FT4 (rs223554 in *DIO1*) were checked as positive controls to ensure the correct use of genomic build (GRCh37), and consistency regarding effect direction and sample size in each study.

Genetic variants with a MAF ≤ 0.5% or poor imputation quality score ≤0.4 were excluded prior to the meta-analyses. Meta-analyses were conducted by two independent analysts using the inverse variance weighting and fixed-effect model approach from the METAL package. Genomic control correction of the individual studies GWAS results was applied if the genomic control parameter lambda (λ GC) was >1.0. Mean numbers of imputed or genotyped variants (autosomal and X-chromosomal) available for the discovery phase were 7,888,577 variants (range 7,771,470–7,987,792) for the continuous traits and 8,146,428 variants (range 8,107,676 – 8,185,180) for the dichotomous traits. Only variants with a MAF > 1% and presence of association in at least 75% of the total sample size (for autosomal and X-chromosomal analyses separately) were considered for further analyses. A pre-specified genome-wide significance threshold of $p$-value of $<5 \times 10^{-8}$ was used, corresponding to a Bonferroni correction for one million independent tests[70]. All reported $p$-values are two-sided, unless stated otherwise. To assess between-study heterogeneity, the $I^2$ statistic was used. Residual inflation of the meta-analysis results was assessed by LD Score Regression intercept using the European ancestry reference panel[71].

To identify independent associated variants within each locus, linkage disequilibrium (LD)-based clumping was performed in PLINK v2.0 using HRC imputed UK Biobank (UKBB) data of European ancestry as a reference ($n = 14$k)[72]. Significance threshold filters (p1 and p2) were set to $5 \times 10^{-8}$ and $r^2$ to ≤0.01 within windows of ±1 Mb. Additionally, independent variants were assessed using the GCTA cojo-slct method with the same LD dataset[73,74]. Significantly associated independent variants with a distance <1MB were combined into a single locus. Variants were considered known if correlated ($R^2 > 0.1$) with a previously known variant within a ±10 Mb distance[24,27]. Loci were considered as known if they include a known variant. The nearest gene around the lead variant was assigned as the locus name. All genomic positions were based on build GRCh37. We used the results of the autoimmune thyroid disease GWAS (30,234 cases and 725,172 controls) for a look-up of all the TSH-associated, high and low TSH loci or their proxies ($r^2 > 0.8$ in a 1 Mb window) that were available in that dataset to assess their relation to autoimmune thyroid disease[50]. Genetic correlation across all traits was assessed using LDscore regression[75]. The variance explained by the significant independent

variants was estimated as the sum of $\beta^2 *2*MAF*(1-MAF)/SD^2$, with $\beta$ representing the variant effect from the meta-analysis and SD the standard deviation of the outcome. For the inverse normal transformed traits, SD equals 1, for the log(FT3/FT4) ratio it was estimated as 0.21 from the two SHIP studies, and for the log(TT3/FT4) ratio it was 0.23 based on the Nijmegen Biomedical Study.

## Fine-mapping

In order to identify causal variants at the associated loci, we applied fine-mapping method SuSiE[76] to the meta-analyses results of each trait. The genomic regions entered into SuSiE were defined as the minimum and maximum positions of each locus plus 500 KB flankings. The same genotype data of 14k UKBB individuals used in LD-based clumping and GCTA COJO analysis were used as reference panel for LD matrix calculation. The R package susieR (version 0.12.35) was used with var_y set to 1, MIN_ABS_CORR to 0.1 and max_iter to 100,000 in susieR functions. All other parameters were set at default. Three loci located in the MHC regions were excluded due to complex LD patterns, and they are for the traits TSH, TT3/FT4 ratio and high TSH. Credible set SNPs and their posterior inclusion probabilities were extracted from susieR outputs. Three loci failed in the susieR run, therefore no credible set could be generated, namely locus 9 in FT4, locus 3 in TT3 and locus 7 in TT3/FT4 ratio. In addition, for locus 3 in TSH susieR gave suspicious results with 10 credible sets (the maximum number of credible sets), and therefore was removed from the final results.

## Colocalization

Colocalization analyses were performed for all genome-wide genetic associations of all eight traits. Gene expression data from 49 tissues in European ancestry samples included in the GTEx Project version 8 release were used (https://gtexportal.org/). We intentionally did not limit our analyses to tissues involved in the hypothalamus-pituitary-thyroid axis, as circulating thyroid hormone concentrations are determined by various processes such as transport and metabolism which take place in peripheral tissues. Both expression quantitative trait loci (eQTL) and GWAS effect alleles and betas were aligned. Genes within a 1.1 Mb region around an independent significant GWAS variant were considered for the colocalization analysis. GWAS results within a ± 100 kb region of each independent significant GWAS variant were extracted, merged with corresponding GTEx eQTL result per tissue, and used as input data for the corresponding analysis. The final colocalization analyses were run using the coloc.fast function of the R-package gtx version 2.1.6 (https://github.com/tobyjohnson/gtx) with default parameters, which includes an adapted implementation of the Giambartolomei's colocalization method. For all colocalization analyses a posterior probability of ≥ 0.85 of the H4 hypothesis (the probability that the assumed model is true and both trait and expression data are associated and share the same single causal variant) was applied to identify significant results.

## Pathway analysis

To elucidate the underlying molecular mechanisms determining thyroid function, the most likely causal genes of the thyroid function traits were prioritized using Data-driven Expression-Prioritized Integration for Complex Traits (DEPICT; version 1 rel194)[77]. In total 296 genes were included for TSH, 128 genes for FT4, 35 for FT3/FT4 ratio and 41 for low TSH, as multiple genes per loci can be assigned in DEPICT. T3, TT3/FT4 ratio and high TSH were not included in the analysis because of the absence of chromosome X data availability in DEPICT and/or the presence of too few significant associations per trait (n < 10). GWAS significant variants were included in the analyses and the following parameters were passed to DEPICT: based on 500 null GWAS the false discovery rate was calculated using 50 repetitions and 500 permutations were used to adjust for potential biases such as gene length. The pathway analyses (core analysis) were generated through

the use of QIAGEN Ingenuity Pathway Analysis (IPA) (QIAGEN Inc., Redwood City, CA, USA)[33]. The genes derived from DEPICT were used as input for the Ingenuity Pathway Analysis to assign canonical pathways (Supplementary Data 15). Direct and indirect relationships were considered in the IPA Knowledge Base reference set for mammals for which the confidence was experimentally observed or high (predicted). Networks were generated with allowing up to 35 molecules/genes per network and a maximum of 25 networks per analysis. No restrictions were applied for tissue and cell lines or mutation. The networks in the analyses were built based on the IPA algorithm which generates a score. The score is based on p-value calculations, reflecting the likelihood of the network molecules of the user-defined set of focus genes being found together by random chance. P-values were calculated using the right-tailed fisher exact test and corrected for multiple testing using the Benjamini-Hochberg correction integrated in the tool[78].

### Tissue expression analysis

For performing gene-set and tissue enrichment analyses, the MAGMA software implemented in the FUMA platform (v1.3.7) was used for 54 tissue types based on GTEx v8 RNA-seq data. MAGMA gene-property test was performed for average gene expression per category (tissue type conditioning on average expression across all categories (oneside)). A Bonferroni correction for multiple testing was performed (p-value = 0.05/54 tissues). This tests a positive relationship between gene expression in a specific category and genetic associations. The specific analysis method of the MAGMA gene property analysis are described in detail elsewhere[41,79].

### Thyroid single-cell RNA sequencing based analyses

To identify key cell types contributing to genetic associations of thyroid function traits, we performed heritability enrichment analysis using single-cell RNA sequencing data from healthy thyroid tissues[42]. This dataset included 54,726 cells in 8 cell clusters and was downloaded from GEO (accession number GSE182416). Mitochondria genes and genes with zero counts in all cells were removed. The LD score regression for specifically expressed genes method was applied to the dataset[80]. The specifically expressed genes in each of the eight thyroid cell types were derived from t-statistics of a linear model fit. The top 10% specifically expressed genes were feed into LDSC software together with thyroid function GWAS summary statistics. We performed the analysis for GWAS of TSH, low TSH, high TSH, FT3, FT4, and FT3/FT4 ratio.

Using LDSC software, annotation files using the specifically expressed genes were created. An annotation file using all genes in the dataset was also generated as a control annotation. For each annotation, the LD scores were then calculated. Per-SNP heritability (h2) enrichment was tested for each annotation using all genes' annotation and baseline model v1.1 annotations as covariants. The coefficient p-value was calculated from the coefficient (tau) z-score[80]. Bonferroni correction was applied to account for the testing of eight cell types.

### Causal associations with thyroid function-related outcomes

To examine the causal association between thyroid function parameters (exposure) and various clinically related traits, two-sample MR was performed using the R-package TwoSampleMR, which facilitates the use of SNP-outcome and SNP-exposure associations obtained from distinct GWAS datasets to assess causality. Based on clinical relevance, the tested clinical outcomes included: body mass index, height, waist-hip ratio, heart rate (resting state), blood pressure (systolic/diastolic), pulse pressure, lipids (LDL, HDL, total cholesterol, triglycerides), type 2 diabetes mellitus, atrial fibrillation, coronary artery disease, heart failure, stroke, bone mineral density, fractures, muscle weakness, Intelligence Quotient, Alzheimer's disease, major depressive disorder,

anxiety and bipolar disorder. For all of these outcomes, GWAS summary statistics from the largest published studies at the time of conducting this study were used[81–97]. Due to large heterogeneity observed in the inverse variance weighted (IVW) analysis[98], the weighted median[99] with its assumption that the majority of the included instruments are valid, was used as the main analysis. Sensitivity analyses included: the IVW that combines the ratio estimators of the individual instruments by a random-effects inverse variance weighted meta-analysis; the heterogeneity-robust Egger regression[100]; the conservative weighted mode[101], and the MR-PRESSO[102] correcting for outliers. A Bonferroni correction for the 24 traits (p-value < 0.05/24 = 0.002) was applied on the weighted median p-value to define significant MR results. Chromosome X was not assessed in the majority of the GWAS used for lookup. Thus, TT3 and TT3/FT4 ratio were excluded from all MR analyses because of the low number of independent significant autosomal variants (n = 2 and n = 6, respectively).

### Pleiotropic effects of thyroid function parameters on clinical endpoints

Phenome-wide association (PheWAS) analyses were performed in order to identify pleiotropic effects of the different thyroid traits PGSs with 1460 diseases from the UK Biobank (n = 379,640) assessed via Hospital Episode Statistics (HES)[103]. HES is a database containing details of all admissions at NHS hospitals in England. An unweighted PGS of each thyroid function trait was used as exposure. Additional details regarding the methodology can be found elsewhere[104]. Associations were reported when passing Bonferroni correction for the number of outcomes tested (p-value < 0.05/1460, thus p-value < 3.4 × $10^{-5}$). For similar reasons mentioned above, TT3 and TT3/FT4 ratio were left out of this analysis.

### Polygenic score analysis on thyroid cancer

The risk association of thyroid cancer attributed to the genome-wide significant variants associated with thyroid function was estimated. For this, a weighted PGS with the effects of each variant aligned to the thyroid trait-increasing allele was created in each individual of the Icelandic deCODE cohort that was not part of our GWAS meta-analysis. The PGS was normalized to a range of 0 to 100 and associated with non-medullary (papillary or follicular) thyroid cancer in up to 792 cases and 148,843 controls either continuously or after binning into quartiles using logistic regression adjusted for sex and age. The probability of disease was calculated using the formula $1/(1 + \exp(-(\beta_0 + \beta_1*x)))$, where $\beta_0$ and $\beta_1$ correspond to the intercept and PGS-related effect in the unadjusted regression model, respectively. TT3 and TT3/FT4 were excluded from the analyses.

### Mendelian randomization on thyroid cancer

The causal effects of the thyroid function parameters on thyroid cancer were assessed by performing two-sample MR analyses. Summary data for the outcome were derived from a meta-analysis of non-medullary thyroid cancer from five cohorts (n = 3100 cases, 287,550 controls) from European descent, including Iceland, Columbus (USA), Houston (USA), Nijmegen (The Netherlands), and Zaragoza (Spain)[105]. Of the total 423 distinct variants from the examined traits, 66 variants were excluded from the MR analysis due to the unavailability of the specific variant in the thyroid cancer GWAS or after variant harmonization (Supplementary Data 17). TT3 (n = 2 variants) and TT3/FT4 ratio (n = 4 variants) were not included the analyses due to limited available variants. We employed the IVW MR as main analysis, and MR-Egger, weighted median and MR-PRESSO as sensitivity analyses. Calculations were performed using the R package TwoSampleMR with default options for data preprocessing and variant harmonization, but without additional LD-filtering[106]. Significance was set to 0.05/6 = 0.008 correcting for the number of exposures tested.

**Reporting summary**

Further information on research design is available in the Nature Portfolio Reporting Summary linked to this article.

## Data availability

The individual participant data included in this project are generally not publicly available due to data privacy laws, but can be applied from the individual studies on reasonable request. We reused publically available data from the GTEx Project version 8 release (https://gtexportal.org/), the GEO accession number GSE182416 (https://www.ncbi.nlm.nih.gov/geo/query/acc.cgi?acc=GSE182416), as well as summary statistics of the following GWAS: body mass index, height, waist-hip ratio, heart rate (resting state), blood pressure (systolic/diastolic), pulse pressure, lipids (LDL, HDL, total cholesterol, triglycerides), type 2 diabetes mellitus, atrial fibrillation, coronary artery disease, heart failure, stroke, bone mineral density, fractures, muscle weakness, Intelligence Quotient, Alzheimer's disease, major depressive disorder, anxiety, bipolar disorder, thyroid cancer, and thyroid function (TSH and FT4), with corresponding references for access provided in the Methods section. The data of the UK Biobank can be applied via the study website (https://www.ukbiobank.ac.uk/). The summary statistics from the GWAS meta-analyses as well as the complete colocalization results and regional association plots of the fine-mapping analyses generated in this project are available on the ThyroidOmics Consortium website (http://www.thyroidomics.com) at the Datasets section (https://transfer.sysepi.medizin.uni-greifswald.de/thyroidomics/datasets/). Source data are provided with this paper.

## Code availability

Unless stated otherwise, GWAS QC, post processing and analyses were implemented in linux shell, perl v5, python v2.7, and R v3.6 using the packages gtx (https://github.com/tobyjohnson/gtx), ggplot2, EasyQC, and TwoSampleMR. Additional analysis software used includes LD Score Regression (https://github.com/bulik/ldsc), FUMA (https://fuma.ctglab.nl/), METAL (www.sph.umich.edu/csg/abecasis/metal/), PLINK (https://www.cog-genomics.org/plink2/), DEPICT (https://github.com/perslab/depict), IPA (https://digitalinsights.qiagen.com/IPA), and GCTA (https://yanglab.westlake.edu.cn/software/gcta/).

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

## Acknowledgements

This work was supported by funding from the European and American Thyroid Associations, the Erasmus University Rotterdam, the Dutch Organization for Scientific Research (NWO) (M.Med.), and the NIH (grants R35GM118335 and T32DK110966). Acknowledgments and study-specific acknowledgments are provided in the Supplementary Note. We conducted this research using the UK Biobank resource under the application numbers 53723 and 20272.

## Author contributions

Project design, analysis or interpretation of the results: A.Kö., A.T., E.M., I.S., J.W.S., M.Med., M.Bu., R.B.S., R.P.P., T.E.G., T.N., W.E.V., Y.L. Drafting of manuscript: A.T., E.M., I.S., M.Med., M.Bu., R.B.S., T.N., Y.L. Design, management or subject recruitment of the individual studies: A.A., A.Kj., A.Kö., A.L., A.M., A.Pe., A.Pu., A.R.C., A.T., B.D.M., B.H.W., B.L., B.M.P., B.S., C.E., C.F., C.G., C.H., C.L., C.Pal., C.Pat., C.V., D.I.C., D.J.S., D.v.H., E.K., E.S., E.S-P., F.C., G.C., G.L., H.V., I.J.D., I.M., J.B.E.T., J.F., J.G.E., J.K.K., J.Lu., J.M.S., J.P.W., J.W.J., K.A.R., L.Ki., L.F., L.Kå., L.T.M., M.Be., M.G., M.Wr., M.K., M.K.S., M.Med., M.Mei., M.M.v.d.K., M.N., M.P.C., M.Wa., N.G., N.G.M., N.J.W., O.Pe., O.Po., P.E.S., P.Re., P.Ri., P.P., R.B.S., R.N-M, R.P.P., S.Ba., S.Wi., S.M., S.N., S.T., S.Z., T.C.R., T.Sp., T.E.G., T.H., T.So., T.Z., T.v.d.E., U.T.S., V.T., W.E.V., W.Z., Y.O. Genotyping or phenotyping of the individual studies: A.G.U., A.T., C.F., C.G., C.N., D.I.C., E.K., F.G., I.J.D., I.M., J-B-J., J.B.R., J.G.E., J.H., J.Lu., J.Be., J.P.W., J.V.L., J.W.J., L.A., L.Ki., M.S.O., N.G., N.G.M., R.C.d.A., S.Har., S.Wi., S.M.C., S.T., S.We., T.Sp., T.H., T.M. Statistical methods, analysis, bioinformatics or interpretation of the results in the individual studies: A.A., A.Kj., A.Kö., A.L., A.M., A.Pa., A.Pu., A.R.C., A.T., B.B., B.D.M., B.H.W., B.K., B.L., B.M.P., B.N.H., B.O.Å., C.E., C.F., C.H., C.J.W., C.N., C.Pal., C.S., C.V., D.C.L., D.I.C., D.R., D.V., D.v.H., E.B.v.d.A., E.K., E.M., F.G., G.L., G.Z., H.J.W., I.G., I.M., I.S., J.Br., J.B-J., J.F., J.G.E., J.K.K., J.La., J.Lu., J.P.W., J.T., J.V.L., K.A.R., L.A., L.C., L.Kå., L.T.M., M.Be., M.K.S., M.Med., M.M.v.d.K., M.Bu., M.P., M.S.O., N.G., O.B., O.Pe., O.Po., P.E.S., P.J.C., P.Ri., Q.Y., R.B.S., R.C.d.A., R.N., S.E.G., S.Har., S. Wi., S.Hau., S.Br., S.L., S.M.C., S.N., S.O., S.S., S.T., T.B., T.E.G., T.H., T.I., T.So., T.N., T.T., T.Z., T.v.d.E., V.T., W.Z. Critical review of manuscript: all authors.

## Competing interests

B.M.P. serves on the Steering Committee of the Yale Open Data Access Project funded by Johnson & Johnson not directly related to this project. The remaining authors declare no competing interests.

## Additional information

Rosalie B. T. M. Sterenborg [1,2,110], Inga Steinbrenner [3,110], Yong Li [3,110], Melissa N. Bujnis[4,110], Tatsuhiko Naito [5,6,110], Eirini Marouli [7,8,110], Tessel E. Galesloot[9], Oladapo Babajide[7], Laura Andreasen[10,11], Arne Astrup [12], Bjørn Olav Åsvold[13,14], Stefania Bandinelli[15], Marian Beekman [16], John P. Beilby [17], Jette Bork-Jensen[18], Thibaud Boutin[19], Jennifer A. Brody [20], Suzanne J. Brown [21], Ben Brumpton [13,22], Purdey J. Campbell[21], Anne R. Cappola[23], Graziano Ceresini[24,25], Layal Chaker[26,27,28], Daniel I. Chasman [29,30], Maria Pina Concas [31], Rodrigo Coutinho de Almeida[16], Simone M. Cross[32], Francesco Cucca[33,34], Ian J. Deary [35], Alisa Devedzic Kjaergaard [36], Justin B. Echouffo Tcheugui[37], Christina Ellervik [30,38,39,40], Johan G. Eriksson[41,42], Luigi Ferrucci [43], Jan Freudenberg[44], GHS DiscovEHR*, Regeneron Genetics Center*, Christian Fuchsberger[45], Christian Gieger [46], Franco Giulianini[29], Martin Gögele[45], Sarah E. Graham [47], Niels Grarup [18], Ivana Gunjača [48], Torben Hansen [18], Barbara N. Harding[20,49], Sarah E. Harris [35], Stig Haunsø[10,50], Caroline Hayward[19], Jennie Hui[51,52], Till Ittermann[53,54], J. Wouter Jukema [55,56], Eero Kajantie [57,58,59], Jørgen K. Kanters [11,60], Line L. Kårhus[61], Lambertus A. L. M. Kiemeney [9,62], Margreet Kloppenburg[63], Brigitte Kühnel[46], Jari Lahti [64], Claudia Langenberg[65,66,67], Bruno Lapauw[68], Graham Leese [69], Shuo Li [70], David C. M. Liewald [35], Allan Linneberg [60,71], Jesus V. T. Lominchar[18], Jian'an Luan [65], Nicholas G. Martin [32], Antonela Matana[48], Marcel E. Meima[2], Thomas Meitinger [72], Ingrid Meulenbelt [16], Braxton D. Mitchell [73,74], Line T. Møllehave[61], Samia Mora [29,30], Silvia Naitza[33], Matthias Nauck [54,75], Romana T. Netea-Maier [1], Raymond Noordam [76], Casia Nursyifa[18], Yukinori Okada [5,6,77,78,79], Stefano Onano [33], Areti Papadopoulou [7], Colin N. A. Palmer [80], Cristian Pattaro[45], Oluf Pedersen [18,81], Annette Peters [82,83], Maik Pietzner [65,66,67], Ozren Polašek[84,85], Peter P. Pramstaller[45], Bruce M. Psaty [20,86], Ante Punda[87], Debashree Ray [88], Paul Redmond[35], J. Brent Richards [89], Paul M. Ridker [29,30], Tom C. Russ [35,90], Kathleen A. Ryan [73], Morten Salling Olesen [10,11], Ulla T. Schultheiss [3,91], Elizabeth Selvin [88], Moneeza K. Siddiqui [92], Carlo Sidore [33], P. Eline Slagboom [16], Thorkild I. A. Sørensen [18,93], Enrique Soto-Pedre[80], Tim D. Spector [94], Beatrice Spedicati [31,95], Sundararajan Srinivasan[80], John M. Starr[90], David J. Stott[96], Toshiko Tanaka[43], Vesela Torlak[87], Stella Trompet [55,76], Johanna Tuhkanen[64], André G. Uitterlinden [26], Erik B. van den Akker [16,97,98], Tibbert van den Eynde[67], Melanie M. van der Klauw [99], Diana van Heemst[76], Charlotte Verroken[68], W. Edward Visser [2], Dina Vojinovic[16,27], Henry Völzke[53,54], Melanie Waldenberger [46], John P. Walsh[21,100], Nicholas J. Wareham [65], Stefan Weiss [54,101], Cristen J. Willer [47], Scott G. Wilson [17,21,94], Bruce H. R. Wolffenbuttel [99], Hanneke J. C. M. Wouters[99], Margaret J. Wright [102], Qiong Yang [70], Tatijana Zemunik [48,87], Wei Zhou [103,104], Gu Zhu[32], Sebastian Zöllner [105,106], Johannes W. A. Smit[1], Robin P. Peeters[2], Anna Köttgen [3,88,107], Alexander Teumer [53,54,108,109,111] ✉ & Marco Medici [1,2,111] ✉

[1]Department of Internal Medicine, Division of Endocrinology, Radboud University Medical Center, Nijmegen, The Netherlands. [2]Academic Center for Thyroid Diseases, Department of Internal Medicine, Erasmus Medical Center, Rotterdam, The Netherlands. [3]Institute of Genetic Epidemiology, Faculty of Medicine and Medical Center - University of Freiburg, Freiburg, Germany. [4]The University of Utah, Salt Lake City, UT, USA. [5]Department of Statistical Genetics, Osaka University Graduate School of Medicine, Suita, Japan. [6]Laboratory for Systems Genetics, RIKEN Center for Integrative Medical Sciences, Kanagawa, Japan. [7]William Harvey Research Institute, Barts and The London School of Medicine and Dentistry, Queen Mary University of London, London, United Kingdom.

[8]Digital Environment Research Institute, Queen Mary University of London, London, UK. [9]Department for Health Evidence, Radboud University Medical Center, Nijmegen, The Netherlands. [10]Laboratory for Molecular Cardiology, Department of Cardiology, Copenhagen University Hospital - Rigshospitalet, Copenhagen, Denmark. [11]Department of Biomedical Sciences, University of Copenhagen, Copenhagen, Denmark. [12]Department of Obesity and Nutritional Sciences, The Novo Nordisk Foundation, Hellerup, Denmark. [13]K.G. Jebsen Center for Genetic Epidemiology, Department of Public Health and Nursing, NTNU, Norwegian University of Science and Technology, Trondheim, Norway. [14]Department of Endocrinology, Clinic of Medicine, St. Olavs Hospital, Trondheim University Hospital, Trondheim, Norway. [15]Geriatric Unit, Azienda Sanitaria Toscana Centro, Florence, Italy. [16]Department of Biomedical Data Sciences, Section Molecular Epidemiology, Leiden University Medical Center, Leiden, The Netherlands. [17]School of Biomedical Sciences, The University of Western Australia, Perth, WA 6009, Australia. [18]Novo Nordisk Foundation Center for Basic Metabolic Research, Faculty of Health and Medical Sciences, University of Copenhagen, Copenhagen, Denmark. [19]MRC Human Genetics Unit, Institute of Genetics and Cancer, University of Edinburgh, Western General Hospital, Edinburgh, United Kingdom. [20]Cardiovascular Health Research Unit, Department of Medicine, University of Washington, Seattle, WA, USA. [21]Department of Endocrinology and Diabetes, Sir Charles Gairdner Hospital, Nedlands, WA 6009, Australia. [22]HUNT Research Centre, Department of Public Health and Nursing, NTNU, Norwegian University of Science and Technology, Levanger 7600, Norway. [23]Division of Endocrinology, Diabetes, and Metabolism, University of Pennsylvania, Philadelphia, PA, USA. [24]Oncological Endocrinology, University of Parma, Parma, Italy. [25]Azienda Ospedaliero-Universitaria di Parma, Parma, Italy. [26]Department of Internal Medicine, Erasmus Medical Center, Rotterdam, The Netherlands. [27]Department of Epidemiology, Erasmus MC, University Medical Centre, Rotterdam, The Netherlands. [28]Department of Epidemiology, Harvard T.H. Chan School of Public Health, Boston, MA, USA. [29]Division of Preventive Medicine, Brigham and Women's Hospital, Boston, USA. [30]Harvard Medical School, Boston, USA. [31]Institute for Maternal and Child Health – IRCCS "Burlo Garofolo", Trieste, Italy. [32]QIMR Berghofer Medical Research Institute, Brisbane, QLD, Australia. [33]Istituto di Ricerca Genetica e Biomedica, Consiglio Nazionale delle Ricerche, 09042 Monserrato (CA), Italy. [34]Università di Sassari, Dipartimento di Scienze Biomediche, V.le San Pietro, 07100 Sassari (SS), Italy. [35]Lothian Birth Cohorts, Department of Psychology, University of Edinburgh, EH8 9JZ Edinburgh, United Kingdom. [36]Steno Diabetes Center Aarhus, Aarhus University Hospital, Palle Juul-Jensens Blvd. 11, Entrance A, 8200 Aarhus, Denmark. [37]Division of Endocrinology, Diabetes, and Metabolism, Johns Hopkins School of Medicine, Baltimore, MD 21205, USA. [38]Faculty of Medical Science, Department of Clinical Medicine, University of Copenhagen, Copenhagen, Denmark. [39]Department of Laboratory Medicine, Boston Children's Hospital, Boston, MA, USA. [40]Department of Clinical Biochemistry, Zealand University Hospital, Køge, Denmark. [41]Department of General Practice and Primary health Care, University of Helsinki, Helsinki, Finland. [42]National University Singapore, Yong Loo Lin School of Medicine, Department of Obstetrics and Gynecology, Singapore, Singapore. [43]Longitudinal Study Section, National Institute on Aging, Baltimore, MD, USA. [44]Regeneron Pharmaceuticals, Inc., Tarrytown, New York, USA. [45]Institute for Biomedicine (affiliated with the University of Lübeck), Eurac Research, Bolzano, Italy. [46]Research Unit Molecular Epidemiology, Institute of Epidemiology, Helmholtz Zentrum München, Neuherberg, Germany. [47]Department of Internal Medicine, Cardiology, University of Michigan, Ann Arbor, MI 48109, USA. [48]Department of Medical Biology, University of Split, School of Medicine, Split, Croatia. [49]Barcelona Institute for Global Health, Barcelona, Spain. [50]Department of Clinical Medicine, University of Copenhagen, Copenhagen, Denmark. [51]Pathwest Laboratory Medicine WA, Nedlands, WA 6009, Australia. [52]School of Population and Global Health, The University of Western Australia, Crawley, WA 6009, Australia. [53]Institute for Community Medicine, University Medicine Greifswald, 17475 Greifswald, Germany. [54]DZHK (German Center for Cardiovascular Research), partner site Greifswald, Greifswald, Germany. [55]Department of Cardiology, Leiden University Medical Center, Leiden, the Netherlands. [56]Netherlands Heart Institute, Utrecht, the Netherlands. [57]Finnish Institute for Health and Welfare, Population Health Unit, Helsinki and Oulu, Oulu, Finland. [58]Clinical Medicine Research Unit, MRC Oulu, Oulu University Hospital and University of Oulu, Oulu, Finland. [59]Department of Clinical and Molecular Medicine, Norwegian University of Science and Technology, Trondheim, Norway. [60]Center of Physiological Research, University of California San Francisco, San Francisco, USA. [61]Center for Clinical Research and Prevention, Bispebjerg and Frederiksberg Hospital, Copenhagen, Denmark. [62]Department of Urology, Radboud University Medical Center, Nijmegen, The Netherlands. [63]Departments of Rheumatology and Clinical Epidemiology, Leiden University Medical Center, Leiden, The Netherlands. [64]Department of Psychology and Logopedics, Faculty of Medicine, University of Helsinki, Helsinki, Finland. [65]MRC Epidemiology Unit, Institute of Metabolic Science, University of Cambridge School of Clinical Medicine, Cambridge CB2 0QQ, UK. [66]Computational Medicine, Berlin Institute of Health at Charité – Universitätsmedizin Berlin, Berlin, Germany. [67]Precision Healthcare University Research Institute, Queen Mary University of London, London, UK. [68]Department of Endocrinology, Ghent University Hospital, C. Heymanslaan 10, 9000 Ghent, Belgium. [69]NHS Tayside, Scotland Dundee UK. [70]Department of Biostatistics, Boston University, Boston, MA, USA. [71]Department of Clinical Medicine, Faculty of Health and Medical Sciences, University of Copenhagen, Copenhagen, Denmark. [72]Institute for Human Genetics, Technical University of Munich, Munich, Germany. [73]University of Maryland School of Medicine, Division of Endocrinology, Diabetes and Nutrition, Baltimore, USA. [74]Geriatrics Research and Education Clinical Center, Baltimore Veterans Administration Medical Center, Baltimore, MD 21201, USA. [75]Institute of Clinical Chemistry and Laboratory Medicine, University Medicine Greifswald, Greifswald, Germany. [76]Department of Internal Medicine, Section of Gerontology and Geriatrics, Leiden University Medical Center, Leiden, the Netherlands. [77]Department of Genome Informatics, Graduate School of Medicine, the University of Tokyo, Tokyo, Japan. [78]Laboratory of Statistical Immunology, Immunology Frontier Research Center (WPI-IFReC), Osaka University, Suita, Japan. [79]Premium Research Institute for Human Metaverse Medicine (WPI-PRIMe), Osaka University, Suita, Japan. [80]Division of Population Health Genomics, School of Medicine, University of Dundee, DD19SY Dundee, UK. [81]Center for Clinical Metabolic Research, Herlev-Gentofte University Hospital, Copenhagen, Denmark. [82]Institute of Epidemiology, Helmholtz Zentrum München, German Research Center for Environmental Health, Neuherberg, Germany. [83]Chair of Epidemiology, Institute for Medical Information Processing, Biometry and Epidemiology, Medical Faculty, Ludwig-Maximilians-Universität München, Munich, Germany. [84]Department of Public Health, University of Split, School of Medicine, Split, Croatia. [85]Algebra University College, Zagreb, Croatia. [86]Departments of Epidemiology and Health Systems and Population Health, University of Washington, Seattle, WA, USA. [87]Department of Nuclear Medicine, University Hospital Split, Split, Croatia. [88]Department of Epidemiology, Johns Hopkins Bloomberg School of Public Health, Baltimore, MD 21205, USA. [89]Lady Davis Institute, Jewish General Hospital, Montreal, Quebec H3T 1E2, Canada. [90]Alzheimer Scotland Dementia Research Centre, University of Edinburgh, Edinburgh, United Kingdom. [91]Department of Medicine IV – Nephrology and Primary Care, Faculty of Medicine and Medical Center – University of Freiburg, Freiburg, Germany. [92]Wolfson Institute of Population Health, Queen Mary University of London, London, UK. [93]Department of Public Health, Section of Epidemiology, Faculty of Health and Medical Sciences, University of Copenhagen, Copenhagen, Denmark. [94]The Department of Twin Research & Genetic Epidemiology, King's College London, St Thomas' Campus, Lambeth Palace Road, London SE1 7EH, UK. [95]Department of Medicine, Surgery and Health Sciences, University of Trieste, Trieste, Italy. [96]Institute of Cardiovascular and Medical Sciences, College of Medical, Veterinary and Life Sciences, University of Glasgow, Glasgow, United Kingdom. [97]Leiden Computational Biology Center, Leiden University Medical Center, Leiden, The Netherlands. [98]Department of Pattern Recognition and Bioinformatics, Delft University of Technology, Delft, The Netherlands. [99]Department of Endocrinology, University of Groningen, University Medical Center Groningen, Groningen, The Netherlands. [100]Medical School, The University of Western Australia, Crawley, WA 6009, Australia. [101]Interfaculty Institute for Genetics and Functional Genomics, University Medicine Greifswald, Greifswald, Germany. [102]Queensland Brain Institute, University of Queensland, Brisbane, QLD, Australia. [103]Analytic and Translational Genetics Unit, Massachusetts General Hospital, Boston,

MA, USA. [104]Program in Medical and Population Genetics, Broad Institute of Harvard and MIT, Cambridge, MA, USA. [105]Department of Biostatistics, University of Michigan, Ann Arbor, MI 48109, USA. [106]Department of Psychiatry, University of Michigan, Ann Arbor, MI 48109, USA. [107]CIBSS – Centre for Integrative Biological Signalling Studies, Albert-Ludwigs-Universität Freiburg, Freiburg, Germany. [108]Department of Psychiatry and Psychotherapy, University Medicine Greifswald, Greifswald, Germany. [109]Department of Population Medicine and Lifestyle Diseases Prevention, Medical University of Bialystok, Bialystok, Poland. [110]These authors contributed equally: Rosalie B. T. M. Sterenborg, Inga Steinbrenner, Yong Li, Melissa N. Bujnis, Tatsuhiko Naito, Eirini Marouli. [111]These authors jointly supervised this work: Alexander Teumer, Marco Medici. *A list of authors and their affiliations appears at the end of the paper.
✉e-mail: ateumer@uni-greifswald.de; m.medici@erasmusmc.nl

## GHS DiscovEHR

Jan Freudenberg[44]

## Regeneron Genetics Center

Jan Freudenberg[44]

