## [Peer Review File · Nature Communications]

Multi-trait analysis characterizes the genetics of thyroid function and identifies causal associations with clinical implicationsREVIEWER COMMENTS

Reviewer #1 (Remarks to the Author):

Sterenberg et al. report results of a meta analysis of GWAS of several thyroid traits in up to 270k individuals of European ancestry. This represents a very well powered study. It is well described and the statistical approach is sound. The authors identify novel loci for all of the traits that they considered. They continue with a functional characterisation (eQTL and tissue specific expression) of the loci. They also assess genetic correlation between the traits and other clinically relevant traits and diseases. Using MR analysis they show shared causal variants for several clinical traits (e.g. cardiovascular) that have previously already been identified. The study represents a very valuable resource for genetics of thyroid function and as a starting point for mechanistic follow up studies.

Addressing the following comments and questions might further improve the manuscript:

1. Replication and heterogeneity. The study reports on overlap with previously know loci. It would be great to also show correlation of effect sizes - ideally also for the novel loci, to see at least concordance of effect size/direction in other studies. Please also provide some reults on heterogeneity: is there any heterogeneity of effect sizes between the cohorts?
2. The MR analysis is reported on the trait level. Most readery are probably familiar with MR usually applied to individual loci (genetic instruments) that are associated with both phenotypes. Please provide more self contained description of the methodology that incorporates multiple loci. In particular, please also mention in the results, how are the results aggregated across these loci to obtain a single p-value. Please also add a table detailing for each pair of traits, which were the individual instruments and their effect sizes on each of the traits, so that MR analyses can be repeated.
3. Fine mapping: it would be nice to also provide fine mapping results, for example the 95% credible sets of SNPs for each locus.
4. Additional functional annotations might reveal more insights: for example is epigenetic (open chromatin) and/or single cell data RNA-seq or ATACs-seq data on relevant tissues (thyroid or pituitary gland) also available? This could reveal the cell types that drive the disease associations.
5. Analysis of secondary signals. Currently, independent loci are defined by ld clumping. Conditional analysis (conditioning on the lead SNP) might reveal additional signals.
6. "For the understudied thyroid traits, 9 variants (8 loci) were associated with FT3, 18 (4 loci) with TT3, 18 (13 loci) with FT3/FT4, and 4 (4 loci) with the TT3/FT4 ratio (Figure 1-3 and Supplementary Figures 2-4, Supplementary Tables 2-7)." The number of loci seems seems surprisingly small for the relative large sample sizes (59k, 15k, 51k and 15k). What are the corresponding effect sizes and how many loci would be expected based on power analyses?
7. Data availability: please make the full genome wide summary statistics available - currently I was not able to find them on the website that is indicated in the data availability section.
8. What is the overlap between the loci identified in the "case/control" setting and the ones idetnified as QTL? Why would it be informative / important to do the case/control analysis? Any insights that can be derived specifically from that?
9. How were genes in Supplementary Table 2 and 3 identified? Simply by position closest to the SNP?
10. Enriched canonical pathways: which genes enter into the pathway analysis?

11. Why does the number of available variants change between the different traits (continuous: 7.8 vs dichotomous : 8.1 Mio)?

12. Only variants with a MAF>1% and presence of association in at least 75% of the total sample size (for autosomal and X-chromosomal analyses separately) were considered for further analyses. What does "presence of association in at least 75% of the total sample size" mean in this context?

13. Are the tissue expression associations corrected for multiple testing of many tissues?

Reviewer #2 (Remarks to the Author):

In this study, Sterenborg et al. conducted a genome-wide meta-analysis of thyroid function across 46 studies. They have detected novel genetic associations with different thyroid hormone traits (circulating thyroid-stimulating hormone (TSH), free thyroxine (FT4), free and total triiodothyronine (FT3 and TT3), FT3/FT4 and TT3/FT4 ratios) in a large sample size of up to 271,040 European individuals. The data source can be useful to study the genetic effects on thyroid function, and advance our knowledge of understanding the genetic pathogenesis of relevant diseases. However, the current study did not take full advantage of the large data to provide in-depth interpretation of their results. Presented analyses such as colocalization, pathway enrichment, MR etc. are mostly routine analytical procedures, and the authors mainly reported limited descriptions of their results which hardly draw significant conclusions. The biggest novelty of this study is the findings of novel TSH associated variants, and further explorations of their functional roles are encouraged. A suggestion would be performing additional functional studies to validate the target genes and regulatory role of these loci, which will help complete the findings.

More specific comments are as follows:

1. Redundant information was presented in Fig. 1c and Figs 2 and 3.
2. "Genome-wide significant variants were considered known when in ± 1 Mb region of a previously identified genome-wide variant". This criterion to define novel and known variants is quite cursory. Highlight novel variants rather than the entire loci in Figs. 2 and 3 would be more illustrative.
3. The TSH reference range is different in the included studies. What's the significance of using different reference ranges? And how would that impact the GWAS results? The authors should make that clear.
4. Line 290, "31 variants in 9 known and 19 novel loci for low TSH", there would be a counting mistake. Same mistake was found in lines 283-284 "85 genetic variants with FT4 in 39 known and 45 novel loci". The authors should be more careful with all presented numbers.
5. the colocalization was performed with genetic associations of thyroid function and of gene expression in 49 different tissues. What is the significance of choosing tissues irrelevant to the releasing of these hormones?
6. Was any fine-mapping analysis performed before colocalization?
7. Further exploration of colocalization results is rather limited as only significant colocalization counts were reported.
8. Different significant thresholds (FDR $p < 0.05$ and nominal $p < 0.05$) were used to filter causal associations as shown in the paragraph "Causal associations with thyroid function-related outcomes", which should be corrected to keep consistent. Besides, Bonferroni correction is more suggested than FDR to draw credible results.
9. Detailed sample information of clinical endpoints should be provided, including the corresponding sample sizes and races etc. Besides, SNP inclusion criteria for MR analysis should be clearly described.
10. There is not a "error bar" in fig. 8, while it was defined in the figure legend.
11. Too many unnecessary abbreviations were used, such as "European ancestry (EA)" in line 588. Besides, the manuscript should be improved to avoid mistakes.

Response to reviewers' comments manuscript NCOMMS-23-10106-T "Multi-trait analysis characterizes the genetics of thyroid function and identifies causal associations with clinical implications"

We would like to thank the reviewers for the overall positive evaluation of our work, and for providing additional helpful comments and suggestions, which we address in detail below. Changes in the manuscript are indicated using tracked changes.

REVIEWER COMMENTS

Reviewer #1 (Remarks to the Author):

Sterenberg et al. report results of a meta-analysis of GWAS of several thyroid traits in up to 270k individuals of European ancestry. This represents a very well powered study. It is well described and the statistical approach is sound. The authors identify novel loci for all of the traits that they considered. They continue with a functional characterisation (eQTL and tissue specific expression) of the loci. They also assess genetic correlation between the traits and other clinically relevant traits and diseases. Using MR analysis they show shared causal variants for several clinical traits (e.g. cardiovascular) that have previously already been identified. The study represents a very valuable resource for genetics of thyroid function and as a starting point for mechanistic follow up studies.

Addressing the following comments and questions might further improve the manuscript:

1. Replication and heterogeneity. The study reports on overlap with previously known loci. It would be great to also show correlation of effect sizes - ideally also for the novel loci, to see at least concordance of effect size/direction in other studies. Please also provide some results on heterogeneity: is there any heterogeneity of effect sizes between the cohorts?

We agree that it is important to assess potential heterogeneity. Therefore, we had presented the heterogeneity I^2 measure in the GWAS results (Tables S2-S9). For the continuous traits, the majority of the associations showed no to moderate heterogeneity. Heterogeneity was higher for the two binary trait GWAS (i.e., low and high TSH levels). We added this information now to the Results section (L292-294).

Furthermore, we have now added scatter plots of the TSH and FT4 association results with the results of the formerly published study¹ as Supplementary Figure 5. In addition, full look-up results are presented in Supplementary Table 11. All significant associations of our current study were direction-consistent and had similar effect sizes with the former results supporting the robustness of our findings. This information was added to the Results (L297-299).

2. The MR analysis is reported on the trait level. Most readers are probably familiar with MR usually applied to individual loci (genetic instruments) that are associated with both phenotypes. Please provide more self contained description of the methodology that incorporates multiple loci. In particular, please also mention in the results, how are the results aggregated across these loci to obtain a single p-value. Please also add a table detailing for each pair of traits, which were the individual instruments and their effect sizes on each of the traits, so that MR analyses can be repeated.

Thank you for your suggestions. For the two-sample MR, we have added an additional description of the different applied methods. In contrast to the one-sample MR where measurements of both phenotypes (i.e. exposure and outcome) of the same individuals are required, two-sample MR facilitates the inclusion of genetic associations with the exposure (i.e. instruments) and with the outcome using distinct samples. Depending on the specific MR method, the association results of the individual instruments (ratio estimators) are combined by an inverse variance weighted meta-analysis (IVW), rely on the assumption that the majority of the instruments are valid (weighted median), or are used as basis for a linear regression (MR-Egger). We briefly added this information to the Methods (L728-736) including corresponding references (to keep the Methods section concise). In addition, we added a Supplementary Table 19 with the SNP-outcome associations included in the MR. Together with the corresponding SNP-exposure effect sizes from our thyroid function GWAS provided in Tables S2-S9, all MR analyses should now be reproducible.

3. Fine mapping: it would be nice to also provide fine mapping results, for example the 95% credible sets of SNPs for each locus.

Thank you for this suggestion. We have added the credible sets of the fine-mapping results obtained from the SuSiE method (L302-208, Supplementary Figure 8, and Supplementary Table 13).

4. Additional functional annotations might reveal more insights: for example is epigenetic (open chromatin) and/or single cell data RNA-seq or ATACs-seq data on relevant tissues (thyroid or pituitary gland) also available? This could reveal the cell types that drive the disease associations.

As the thyroid is the most interesting tissue in this context, we searched the literature and found single cell RNA sequencing results based on 54,726 cells from normal thyroid tissue of 7 patients who underwent thyroidectomy². GTEx tissue expression analyses of TSH-associated loci had already shown an enrichment of genes expressed in the thyroid. Next, these single-cell RNA sequencing analyses pinpointed that among various cell types present in the thyroid, the epithelial cell (i.e., the thyrocyte) is the responsible cell type. Indeed, the thyrocyte expresses the TSH-receptor and subsequent TSH signaling cascade, and is responsible for thyroid hormone production. This has now been included in the Methods (L700-715), Results (L371-376) and Discussion (459-463) sections.

5. Analysis of secondary signals. Currently, independent loci are defined by ld clumping. Conditional analysis (conditioning on the lead SNP) might reveal additional signals.

Although conditional analysis in large scale GWAS meta-analysis is technically challenging, the GCTA software provides an approach for conducting such an analysis requiring GWAS meta-analysis summary statistics and individual level genotypes for LD estimation as input^{3,4}. However, the results of this method are somewhat sensitive to the input data, i.e. the LD panel and the precision of the GWAS results as the association p-values are re-calculated. Based on our experience, the clumping method using our conservative parameters provides

robust results for revealing independent signals. Nevertheless, we conducted a GCTA co-joint analysis (Methods L616-618) for our thyroid function traits which are now provided in Supplementary Table 12. As indicated also in that table, there is a strong concordance with the clumping results, which is now also mentioned in the results section (L299-301).

6. "For the understudied thyroid traits, 9 variants (8 loci) were associated with FT3, 18 (4 loci) with TT3, 18 (13 loci) with FT3/FT4, and 4 (4 loci) with the TT3/FT4 ratio (Figure 1-3 and Supplementary Figures 2-4, Supplementary Tables 2-7)." The number of loci seems surprisingly small for the relative large sample sizes (59k, 15k, 51k and 15k). What are the corresponding effect sizes and how many loci would be expected based on power analyses?

The corresponding effect sizes for all results are provided in Supplementary Tables 2-9. We agree that the number of significant associations revealed for the understudied traits is lower than for TSH and FT4 even when using comparable sample sizes (i.e. in former GWAS). For a possible explanation it needs to be taken into account that the number of loci that can be revealed by GWAS depends not only on the sample size but also on the strength of the genetic effect of each variant, and the trait variation including its measurement error. Taking FT3 and TSH as an example, both traits were normalized to standard deviation (SD) units which make the GWAS effect sizes and its standard errors (SE) generally comparable, although the individual SD unit represents different ranges for each trait. The unsigned median effect sizes (SE) of the significant associations were 0.017 (0.003) for TSH and 0.052 (0.006) for FT3. The SEs are in line between both traits given the approximately four times higher sample size for TSH resulting in an expected $\frac{1}{2}$ of the FT3 SE. Furthermore, the higher median effect size of FT3 is in line with the lower sample size resulting in a lower power to detect associations for FT3 compared to TSH. Finally, we compared the effect sizes and SEs and checked them for plausibility across traits and studies to avoid technical errors as part of our meta-analysis QC workflow. Thus, it seems rather likely that FT3 has smaller detectable genetic effects attributed to single variants which can be independent of the overall heritability of the trait. Another explanation could be that the body tries to keep the circulating T3 levels as long as possible on a stable level as T3 is the effective hormone, such as can be seen in hypothyroidism where there is preferential thyroidal T3 production and increased peripheral T4 to T3 conversion. Of note, there are also other traits that have a similar small number of GWAS associations using similar or larger sample sizes (and unlikely explainable only by the imputation reference panel). Blood pressure related measurements are one example⁵.

While working on the revision of our manuscript, we realized that there was a technical error (server problem) in the clumping of the TT3/FT4 ratio results, thereby missing two loci (and their secondary signals). We corrected this error resulting in 18 independent associations in total for this trait. The GWAS meta-analysis results were not affected, only the PheWAS and coloc analyses, which are updated accordingly.

7. Data availability: please make the full genome wide summary statistics available - currently I was not able to find them on the website that is indicated in the data availability section.

We prepared an updated GWAS download page at <https://transfer.sysepi.medizin.uni-greifswald.de/thyroidomics/datasets-new/> which will also be linked to our ThyroidOmics

Consortium website (<http://www.thyroidomics.com>) immediately after acceptance of our manuscript replacing the current GWAS download page.

8. What is the overlap between the loci identified in the "case/control" setting and the ones identified as QTL? Why would it be informative / important to do the case/control analysis? Any insights that can be derived specifically from that?

TSH levels above (subclinical and overt hypothyroidism) and below (subclinical and overt hyperthyroidism) the reference range are associated with an increased risk of adverse outcomes, including atrial fibrillation, coronary heart disease, stroke and mortality. Therefore, the case/control analysis was intended to provide a more clinical assessment regarding high and low TSH values, while the continuous TSH GWAS included only individuals with TSH levels within the reference range. The significantly associated variants emerged in the case/control GWAS could prioritize loci that might be predominantly involved in hypo- and hyperthyroidism rather than in general variation of TSH levels. For example, the rs3130552 near the psoriasis susceptibility 1 candidate 1 (*PSORS1C1*) gene is more strongly associated with clinically high TSH than with TSH in the normal range. *PSORS1C1* is involved in several autoimmune diseases including autoimmune thyroid diseases, as also mentioned in the discussion section (L480).

While all associations with high TSH were also genome-wide significantly associated with variation in reference range TSH levels, three loci for low TSH (*SDCCAG8*, *ZSWIM1* and *DMD*) did not pass this threshold. We added the TSH lookup p-values to Supplementary Tables 8 and 9.

9. How were genes in Supplementary Table 2 and 3 identified? Simply by position closest to the SNP?

Indeed, as written in the Methods the name of the nearest gene around the lead variant was assigned as a locus name. We have updated the column descriptions of the corresponding Supplementary Tables to avoid confusion regarding the naming of a locus. Additionally, we like to stress that the assigned name to a locus is only for convenience and recognition without providing any information of potential causal genes.

10. Enriched canonical pathways: which genes enter into the pathway analysis?

We thank you for this suggestion, and have added the requested information as Supplementary Table 15.

11. Why does the number of available variants change between the different traits (continuous: 7.8 vs dichotomous : 8.1 Mio)?

The difference in number of variants per GWAS trait is attributed to the number of studies and their sample sizes that are finally included in the corresponding meta-analysis.

In detail, there are mainly two aspects that affect the number of variants available in the meta-analysis results. First, the final number of variants that passed imputation QC is specific to each study using different genotyping array types and imputation reference panels. Furthermore, studies may remove variants from their GWAS if association results could not

be calculated, i.e. due to low MAF and low imputation quality resulting in a low effective sample size. Consequently, traits with larger samples sizes in the meta-analysis included more studies, resulting in a higher total number of variants in the meta-analysis result file compared to meta-analysis results with a smaller total sample size (and fewer studies included). This fits nicely the observed relationship between the increasing number of variants at higher sample size in the meta-analysis (the unfiltered METAL results were used to avoid influences by subsequent filters) as shown below.

Of note, the effective sample size of the dichotomous traits depends also on the number of cases, and can thus not directly be compared with a continuous trait of the same total sample size.

trait	# variants	total sample size	# of cohorts
TT3/FT4 ratio	12,862,013	15,510	9
TT3	12,841,651	15,829	9
FT3/FT4 ratio	14,932,191	51,095	21
FT3	15,054,314	59,061	25
low TSH	15,711,308	141,549 (dichotomous)	21
high TSH	15,931,720	153,241 (dichotomous)	27
FT4	16,812,590	119,120	43
TSH	17,199,244	271,040	46

12. Only variants with a MAF>1% and presence of association in at least 75% of the total sample size (for autosomal and X-chromosomal analyses separately) were considered for further analyses. What does "presence of association in at least 75% of the total sample size" mean in this context?

The total sample size that was available in the GWAS meta-analysis varied by each variant mainly depending on the MAF, imputation quality (in combination with the cohort-specific sample size), the cohort specific array type and the imputation reference panel used. For example, smaller cohorts have a higher chance that the association model calculation fails due to lower MAF and imputation quality (resulting in a very low effective sample size). To reduce false positive association results that emerge because of a substantially lower total sample size and power⁶, we filtered such variants after the meta-analysis. This means that we only took forward those variants that showed associations based on a sample size that was minimally 75% of the maximum GWAS meta-analysis sample size. Given that not all cohorts could provide chromosome X GWAS results, we calculated this minimum sample size threshold separately for the autosomes and for chromosome X. Based on former GWAS, setting this cutoff to 25% missingness was an empirical but plausible value.

13. Are the tissue expression associations corrected for multiple testing of many tissues?

Yes, the results of the tissue expression analysis are corrected for multiple testing of the 54 tissues used in GTEx. In the respective Supplementary Figure 9 the dashed line corresponds to the Bonferroni correction ($-\log_{10}(0.05/54)=0.0009$)=3.03). The tissue expression analysis

showed only significant results for TSH, which are depicted in the red bars. We clarified this in the Results section (L367), in the Methods section (L696) and in the description of Supplementary Figure 9.

Reviewer #2 (Remarks to the Author):

In this study, Sterenborg et al. conducted a genome-wide meta-analysis of thyroid function across 46 studies. They have detected novel genetic associations with different thyroid hormone traits (circulating thyroid-stimulating hormone (TSH), free thyroxine (FT4), free and total triiodothyronine (FT3 and TT3), FT3/FT4 and TT3/FT4 ratios) in a large sample size of up to 271,040 European individuals. The data source can be useful to study the genetic effects on thyroid function, and advance our knowledge of understanding the genetic pathogenesis of relevant diseases. However, the current study did not take full advantage of the large data to provide in-depth interpretation of their results. Presented analyses such as colocalization, pathway enrichment, MR etc. are mostly routine analytical procedures, and the authors mainly reported limited descriptions of their results which hardly draw significant conclusions. The biggest novelty of this study is the findings of novel TSH associated variants, and further explorations of their functional roles are encouraged. A suggestion would be performing additional functional studies to validate the target genes and regulatory role of these loci, which will help complete the findings.

Thank you for your thorough and generally positive assessment of our study, and the thoughtful comments. We have taken many of the specific suggestions for adding additional analyses into account, and address them in detail below. Amongst others, this also included analyses using thyroid single cell RNA sequencing results, for which we kindly refer to question 4 of reviewer 1. Unfortunately, experimental studies are beyond the scope of this project, which already combined data from 46 studies, 270,000 individuals, and took almost 4 years to retrieve cohorts with eligible data, perform the study-specific GWAS based on our pre-defined analysis plan, quality control the association results, and conducting the meta-analyses and follow-up analyses. We do make the full genome-wide summary statistics available, to enable all experimental researchers in the thyroid field to use them for experimental follow-up studies. Furthermore, we checked prior to the initial submission of our manuscript for candidates eligible for functional follow-up with our available resources and within a reasonable time frame, but we did not find a possible solution without delaying the paper (and the publication of the GWAS results) for multiple years. We also like to emphasize that besides the finding of additional TSH and FT4 associated hits, we revealed for the first time multiple genetic associations for T3 related traits, and showed the genetic relationships between TSH, FT4 and FT3 which has implications e.g. on Mendelian randomization analyses for assessing causal effects of and on thyroid function.

More specific comments are as follows:

1. Redundant information was presented in Fig. 1c and Figs 2 and 3.

We have moved Figure 1c into a separate Figure (Figure 2) as it provides an overview of the main traits and the overlap of the variants across the traits. In addition, we combined the

former Figures 2 and 3 into a new Figure with two panels (Figure 3). We think that the separate Manhattan plots for TSH and FT4 provide a more detailed overview of the (many) associations compared to the circos plot. However, we have now colored known loci with additional novel independent variants using a separate color (in addition to completely new loci) in the Manhattan plots.

2. “Genome-wide significant variants were considered known when in $\pm 1\text{Mb}$ region of a previously identified genome-wide variant”. This criterion to define novel and known variants is quite cursory. Highlight novel variants rather than the entire loci in Figs. 2 and 3 would be more illustrative.

We agree that the assessment of novel and known loci deserves improvement, and point out that the distinction between loci and individual variants might not have been made fully clear. Now we substantially changed the locus definition, and the assessment of known loci to a variant-based approach using LD information. In detail, significantly associated independent variants with a distance $< 1\text{MB}$ were combined into a locus. Variants were considered as known if they correlate ($R^2 > 0.1$) with a previously known variant within a $\pm 10\text{Mb}$ distance derived from the previous thyroid function GWAS. Loci were considered as known if they include a known variant. We updated the Methods accordingly (L619-621). Of note, we intentionally chose these LD and distance values for assessing known variants in such a way to be rather conservative regarding novel associations.

Furthermore, we updated the (former) Figures 2 and 3 by indicating known loci that include novel independent variants revealed in our current GWAS.

3. The TSH reference range is different in the included studies. What’s the significance of using different reference ranges? And how would that impact the GWAS results? The authors should make that clear.

The major aim of our study is the discovery of genetic factors that determine inter-individual differences in thyroid function, which is assessed best within reference range concentrations (see also Introduction L242). As reference ranges can differ between assays and populations, we asked each cohort to ideally provide the cohort specific reference ranges, defined as the 2.5th and 97.5th percentiles of the TSH distribution in the specific population. This approach takes also additional environmental factors like population iodine supply into account⁷. If this was not possible, the reference range for TSH provided by the assay manufacturer was used. This information has now been added to the Methods (L563-567). After receiving all results, we performed inverse normal transformation to convert the trait to normal distribution for TSH, FT4, FT3 and TT3, and log transformation for the ratios (as the effect size interpretation of an inverse normal transformed ratio is less informative). Thus, the differences in reference ranges used are therefore minor as both approaches removed individuals with untreated hypo- or hyperthyroidism. Treated individuals have been excluded a priori (L554).

4. Line 290, “31 variants in 9 known and 19 novel loci for low TSH”, there would be a counting mistake. Same mistake was found in lines 283-284 “85 genetic variants with FT4 in 39 known and 45 novel loci”. The authors should be more careful with all presented numbers.

We agree that the former representation (and the different definition) of variants and loci could cause confusion. In the revised manuscript we updated both the definition of loci and the assessment of new findings (please see answer to your second question), hoping that it makes the representation and accountability of the presented numbers more suitable for the reader. Of note, given that many genome-wide significant associations in close vicinity are correlated because of their linkage disequilibrium we a) assessed independent associations as significant variants, and b) grouped variants located in close distance into associated loci. This is a common practice in GWAS for presenting the significant results.

5. The colocalization was performed with genetic associations of thyroid function and of gene expression in 49 different tissues. What is the significance of choosing tissues irrelevant to the releasing of these hormones?

The main aim of our GWAS is to identify genetic variants which determine circulating thyroid hormone levels. Next to regulation of thyroid hormone synthesis by the hypothalamus-pituitary-thyroid (HPT) axis, the level of circulating thyroid hormone is determined by various processes taking place in peripheral tissues such as thyroid hormone transport and metabolism. Therefore, our analyses were not limited to the tissues involved in the HPT axis. This is illustrated by one of the FT4 colocalization results including *AADAT* in the small intestine and adipose subcutaneous tissue (rs76767373, rs112649654), which is a known thyroid hormone metabolizing enzyme¹. This has now been discussed in the methods section (L652-655)

6. Was any fine-mapping analysis performed before colocalization?

The colocalization uses all variants of a ± 100 kb region around a GWAS index variant as input, which does not require fine-mapping. We re-wrote the corresponding Methods section (L656-660) to elaborate on this issue. Our fine-mapping was conducted independently of the colocalization analyses.

7. Further exploration of colocalization results is rather limited as only significant colocalization counts were reported.

Colocalization analyses were performed for the significant findings of all thyroid function traits (i.e. more than 400 index variants) with their genes in *cis* across all 49 GTEx tissues. Given the huge number of colocalization results obtained from these analyses (almost 275,000 associations in total), we limited the presentation in the manuscript to the significant ones. However, we now made the full colocalization results available for download together with our GWAS results (please see answer 7 to Reviewer #1) and indicated this also in the Data availability statement (L777).

8. Different significant thresholds (FDR $p < 0.05$ and nominal $p < 0.05$) were used to filter causal associations as shown in the paragraph "Causal associations with thyroid function-related outcomes", which should be corrected to keep consistent. Besides, Bonferroni correction is more suggested than FDR to draw credible results.

In agreement with this suggestion, we applied Bonferroni correction for multiple testing and removed the results that were solely nominally significant from the main text. The additional $p < 0.05$ cut-off was applied in case a reader is interested in a specific trait only (and thus does not need to apply a correction for multiple testing). Given that it is not feasible to illustrate all MR results in a single plot, we decided to keep the nominally significant results in Figure 6.

9. Detailed sample information of clinical endpoints should be provided, including the corresponding sample sizes and races etc. Besides, SNP inclusion criteria for MR analysis should be clearly described.

We agree with this suggestion and provide all MR outcomes including details of the underlying samples in Supplementary Table 18, and all associations of the instruments with the outcomes in Supplementary Table 19. We added this information also to the Results section (L382-383).

10. There is not a “error bar” in fig. 8, while it was defined in the figure legend.

We apologize for this copy-paste error, and removed the sentence from the legend of Figure 8 and (re-numbered) Supplementary Figure 10.

11. Too many unnecessary abbreviations were used, such as “European ancestry (EA)” in line 588. Besides, the manuscript should be improved to avoid mistakes.

Thanks for pointing out this issue. We removed the EA abbreviation from the manuscript, as well as PP, CAD, IQ, BMI and AF. In addition, we corrected a few typos throughout the manuscript.

References

1. Teumer, A. *et al.* Genome-wide analyses identify a role for SLC17A4 and AADAT in thyroid hormone regulation. *Nat Commun* **9**, 4455 (2018).
2. Hong, Y. *et al.* Single Cell Analysis of Human Thyroid Reveals the Transcriptional Signatures of Aging. *Endocrinology* **164**(2023).
3. Yang, J. *et al.* Conditional and joint multiple-SNP analysis of GWAS summary statistics identifies additional variants influencing complex traits. *Nat Genet* **44**, 369-75, S1-3 (2012).
4. Yang, J., Lee, S.H., Goddard, M.E. & Visscher, P.M. GCTA: a tool for genome-wide complex trait analysis. *Am J Hum Genet* **88**, 76-82 (2011).
5. Wain, L.V. *et al.* Genome-wide association study identifies six new loci influencing pulse pressure and mean arterial pressure. *Nat Genet* **43**, 1005-11 (2011).
6. Ioannidis, J.P. Why most discovered true associations are inflated. *Epidemiology* **19**, 640-8 (2008).
7. Volzke, H. *et al.* Reference intervals of serum thyroid function tests in a previously iodine-deficient area. *Thyroid* **15**, 279-85 (2005).

REVIEWER COMMENTS

Reviewer #1 (Remarks to the Author):

I would like to thank the authors for addressing all of my comments. Based on these new results I have one additional question. On the analysis of single cell data the authors report:

"Of the investigated cell types (see Methods), genes identified in the TSH and high TSH GWASs were significantly enriched in thyroid epithelial cells (TSH p-value=0.0004, high TSH p-value=0.0009). No associations with other cell types were observed for the aforementioned and other tested thyroid function related traits".

These results are based on the "coefficient P-value". The supplementary table 17 also reports "Enrichment P". If Bonferroni adjustment is applied on these P-values there are also other cell types that show association with respect to the expression of GWAS genes (like immune cells). I am wondering, what the difference between these two tests is and why to prefer one over the other?

Reviewer #2 (Remarks to the Author):

The authors have addressed a series of mine initial concerns and suggestions. These changes to the manuscript have improved the overall quality of the work. In the revised manuscript, the authors added fine-mapping and single-cell based analyses for further exploration of their GWAS results. However, the interpretation and presentation of these results need to be further improved to show the significance of their dataset.

More detailed comments:

1. What do the authors want to show by estimating the sizes of credible sets resulted from fine-mapping analyses? Besides, the authors have compared the sizes of credible sets among different traits, but how to interpret the differences?
2. In single-cell RNA based analyses, both high TSH and low TSH GWAS hits were enriched in thyroid epithelial cells (the significant result of low TSH was missed in the main text). How to explain this observation?
3. The genetic correlation showed that FT3/FT4 were negatively and positively correlated with FT4 and FT3, respectively, which is expected as the FT3/FT4 ratio is calculated from FT4 and FT3 concentration. Therefore, the conclusion that "FT4 as well as FT3 and TT3 associated loci reflect better the thyroid hormone metabolism" needs additional evidence to support. Besides, it is better to move fig.4 to supplementary file.
4. The authors have identified colocalization in several tissues besides HPT axis. However, the biological explanation and interpretation of these results is rather limited. For instance, the esophagus mucosa and tibial nerve are the second most colocalized tissues for TSH. What's the biological significance? What's the relationship of these tissues with thyroid hormone levels? Rather than displaying the counts of colocalization signals, the authors should better revise fig.5 to express more meaningful information.

Response to reviewers' comments manuscript NCOMMS-23-10106-A "Multi-trait analysis characterizes the genetics of thyroid function and identifies causal associations with clinical implications"

We thank the reviewers for evaluating our revised manuscript, and the positive feedback. We addressed the comments in detail below, and hope that the additional improvements of the manuscript make it suitable for publication. Changes in the manuscript are indicated using tracked changes.

REVIEWER COMMENTS

Reviewer #1 (Remarks to the Author):

I would like to thank the authors for addressing all of my comments. Based on these new results I have one additional question. On the analysis of single cell data the authors report:

"Of the investigated cell types (see Methods), genes identified in the TSH and high TSH GWASs were significantly enriched in thyroid epithelial cells (TSH p-value=0.0004, high TSH p-value=0.0009). No associations with other cell types were observed for the aforementioned and other tested thyroid function related traits".

These results are based on the "coefficient P-value". The supplementary table 17 also reports "Enrichment P". If Bonferroni adjustment is applied on these P-values there are also other cell types that show association with respect to the expression of GWAS genes (like immune cells). I am wondering, what the difference between these two tests is and why to prefer one over the other?

The coefficient represents the increase in per-SNP heritability associated with the given cell type conditional on all cell types and other annotations in the baseline model (including genic regions, enhancer regions, and conserved regions, etc.). The coefficient p-value tests whether the coefficient is significantly positive. In contrast, the enrichment is the marginal increase in per-SNP heritability for SNPs associated with the given cell type to other cell types ($h^2/\%SNPs$), and the enrichment p-value tests if it is significant. For identifying critical disease-relevant tissues and cell types, the coefficient p-value is more appropriate and it is the one used in the paper describing the stratified LD score regression method (Finucane *et al.*, Nat Genet. 2018, PMID: 29632380).

We added the following footnote to Supplementary Table 17:

"The coefficient column represents the increase in per-SNP heritability associated with the given cell type conditional on all cell types and other annotations in the baseline model (including genic regions, enhancer regions, and conserved regions, etc.). The coefficient p-value tests whether the coefficient is significantly positive, and was used to assess tissue enrichment. The enrichment column is the marginal increase in per-SNP heritability for SNPs associated with the given cell type to other cell types ($h^2/\%SNPs$), and the enrichment p-value tests if it is significant."

Reviewer #2 (Remarks to the Author):

The authors have addressed a series of mine initial concerns and suggestions. These changes to the manuscript have improved the overall quality of the work. In the revised manuscript, the authors added fine-mapping and single-cell based analyses for further exploration of their GWAS results. However, the interpretation and presentation of these results need to be further improved to show the significance of their dataset.

More detailed comments:

1. What do the authors want to show by estimating the sizes of credible sets resulted from fine-mapping analyses? Besides, the authors have compared the sizes of credible sets among different traits, but how to interpret the differences?

We wanted to assess whether there are traits that showed a substantially smaller (or larger) average size of credible sets, i.e. that might have an enrichment of potential causal variants among the associated GWAS loci. However, there was no significant difference of the mean size of the credible sets among the eight traits (ANOVA $p=0.97$). We added this information now to the Results (line 303).

2. In single-cell RNA based analyses, both high TSH and low TSH GWAS hits were enriched in thyroid epithelial cells (the significant result of low TSH was missed in the main text). How to explain this observation?

Thank you also for pointing out the issue of the missing low TSH association. Actually, we apologize there was a typo in the main text which probably led to this question. It should state correctly (based on the "Coefficient P" provided in Supplementary Table 17):

"...genes identified in the TSH and low TSH GWASs were significantly enriched in thyroid epithelial cells".

We updated the Results accordingly (line 390). In contrast, the high TSH association did not pass the formal level of significance after Bonferroni correction, i.e. $p = 0.05/8$ cell types = 0.00625.

3. The genetic correlation showed that FT3/FT4 were negatively and positively correlated with FT4 and FT3, respectively, which is expected as the FT3/FT4 ratio is calculated from FT4 and FT3 concentration. Therefore, the conclusion that "FT4 as well as FT3 and TT3 associated loci reflect better the thyroid hormone metabolism" needs additional evidence to support. Besides, it is better to move fig.4 to supplementary file.

We agree that our interpretation of the genetic correlation requires a more detailed explanation:

Based on the significant genetic correlation between TSH and FT3 (genetic correlation = -0.2, FDR <0.001), the associated loci of these traits seem to reflect thyroid function determined by the active thyroid hormone FT3. The non-significant correlation of FT4 with TSH in combination with the significant correlation of FT4 with both ratios (all FDR <0.001) suggest that the FT4 associated loci reflect the thyroid hormone metabolism assessed via the T3/T4

ratio. The significant genetic correlation of TT3 with FT3 (genetic correlation = 0.5, FDR = 0.005) indicates that also the TT3 associated loci predominantly reflect the active thyroid hormone (Figure 4). Taking into account the strong effects of the *SERPINA7* locus for TT3, this could explain the non-significant genetic correlation of TT3 with TSH.

We updated this information in the Results section accordingly (lines 336-344). Taking into account that Figure 4 provides this valuable information, we intend to keep the figure in the main text. However, we updated Figure 4 by removing redundant information and included the genetic correlation values in the lower part.

4. The authors have identified colocalization in several tissues besides HPT axis. However, the biological explanation and interpretation of these results is rather limited. For instance, the esophagus mucosa and tibial nerve are the second most colocalized tissues for TSH. What's the biological significance? What's the relationship of these tissues with thyroid hormone levels? Rather than displaying the counts of colocalization signals, the authors should better revise fig.5 to express more meaningful information.

For TSH, there is a clear enrichment in one tissue (thyroid), followed by many tissues with an at least 5-fold lower number of colocalizations. This is in clear contrast with FT4, in which the thyroid is much less dominantly present compared to other tissues. We believe this is the main message of these analyses, which could (partly) explain the limited overlap between GWAS hits for TSH and the thyroid hormones as discussed in the discussion section (lines 479-490).

We agree that further analyses on the colocalization hits in non-HPT tissues are needed to reveal their connection to thyroid function. For example, peripheral nerve activity is related to thyroid function with more evident abnormalities, i.e. sensorimotor axonal neuropathy, in presence of increased basal TSH levels (Misiunas *et al.* 2015, PMID: 7488869). However, we believe that this type of analysis is beyond the scope of our current manuscript. Therefore, we make all colocalization results publicly available on our consortium website to provide the scientific community the opportunity to follow-up on the tissue of interest.

Based on the Reviewer's suggestions, we added additional plots showing detailed colocalization results with TSH, FT4, and high/low TSH representing the traits with the highest number of colocalizations within the HPT tissues. As seen in Figures 5 (b+c) and the new Supplementary Figure 8, genes like *PDE8B* and *TPO* involved TSH signaling cascade and TH synthesis colocalized in thyroid tissue only. We added this information to the Results (lines 364-366).

REVIEWERS' COMMENTS

Reviewer #2 (Remarks to the Author):

In this revised version the authors have addressed all my concerns which I feel significantly enhance their findings.